# Ventral striatal islands of Calleja neurons bidirectionally mediate depression-like behaviors in mice

Yun-Feng Zhang [1,2,3] ✉, Jialiang Wu[1,2,6], Yingqi Wang[3,6], Natalie L. Johnson[4,6], Janardhan P. Bhattarai[3,6], Guanqing Li[1,2,5], Wenqiang Wang[1,2,5], Camilo Guevara[3], Hannah Shoenhard [3], Marc V. Fuccillo [3], Daniel W. Wesson [4] & Minghong Ma [3] ✉

The ventral striatum is a reward center implicated in the pathophysiology of depression. It contains islands of Calleja, clusters of dopamine D3 receptor-expressing granule cells, predominantly in the olfactory tubercle (OT). These OT D3 neurons regulate self-grooming, a repetitive behavior manifested in affective disorders. Here we show that chronic restraint stress (CRS) induces robust depression-like behaviors in mice and decreases excitability of OT D3 neurons. Ablation or inhibition of these neurons leads to depression-like behaviors, whereas their activation ameliorates CRS-induced depression-like behaviors. Moreover, activation of OT D3 neurons has a rewarding effect, which diminishes when grooming is blocked. Finally, we propose a model that explains how OT D3 neurons may influence dopamine release via synaptic connections with OT spiny projection neurons (SPNs) that project to midbrain dopamine neurons. Our study reveals a crucial role of OT D3 neurons in bidirectionally mediating depression-like behaviors, suggesting a potential therapeutic target.

Depression, one of the most common affective disorders, is associated with dysfunction of brain reward centers[1]. In both rodents[2,3] and humans[4–7], depressive phenotypes are linked to dysfunction of the ventral striatum, a key reward center that integrates brain-wide inputs including from midbrain dopamine neurons and sends inhibitory outputs to downstream structures[8,9]. The ventral striatum contains two major divisions, the nucleus accumbens (NAc) and the olfactory tubercle (OT; also called tubular striatum[10]). Similar to the NAc, the OT is also involved in motivational and reward-related behaviors in rodents[11–18]. In contrast to extensive research on the role of NAc in depression[19–27], whether and how the OT circuitry contributes to depression is largely unknown.

Like the rest of the striatum, the OT contains spiny projection neurons (SPNs) and several types of interneurons[28,29]. SPNs are GABAergic principal neurons that express either D1- or D2-type dopamine receptor[8,9]. Functional and anatomical changes in NAc D1 and D2-SPNs or interneurons have been linked to affective behaviors in rodents[19–27]. The ventral striatum also contains neurons expressing D3-type dopamine receptors, the majority of which are granule cells concentrated in the islands of Calleja of the OT[30–35]. The *drd3* gene encoding the D3 receptor as well as the islands of Calleja have been linked to the pathophysiology of neuropsychiatric diseases[36–41]. Moreover, cariprazine, an atypical antipsychotic drug prescribed to treat several neuropsychiatric disorders including bipolar depression,

[1]State Key Laboratory of Integrated Management of Pest Insects and Rodents, Institute of Zoology, Chinese Academy of Sciences, 100101 Beijing, China. [2]CAS Center for Excellence in Biotic Interactions, University of Chinese Academy of Sciences, 100101 Beijing, China. [3]Department of Neuroscience, University of Pennsylvania Perelman School of Medicine, Philadelphia, PA 19104, USA. [4]Department of Pharmacology and Therapeutics, University of Florida, Gainesville, FL 32610, USA. [5]College of Life Sciences, Hebei University, Baoding 071002 Hebei, China. [6]These authors contributed equally: Jialiang Wu, Yingqi Wang, Natalie L. Johnson, Janardhan P. Bhattarai. ✉e-mail: yfzhang@ioz.ac.cn; minghong@pennmedicine.upenn.edu

predominantly binds to islands of Calleja neurons in the mouse brain[42], supporting a potential role of these neurons in regulating emotion.

We recently discovered that OT D3 neuron activity bidirectionally regulates self-grooming in mice: optogenetic activation of these neurons robustly initiates orofacial grooming while optogenetic inhibition halts ongoing grooming[43]. Self-grooming, an evolutionally-conserved, repetitive behavior, serves important functions including de-arousal and stress reduction[44,45]. Notably, altered levels of grooming are considered a behavioral biomarker for a number of neurological and neuropsychiatric disorders including depression[44,45]. Since self-grooming increases dopamine release in the NAc[46], it is possible that the activity of OT D3 neurons and self-grooming are intimately linked to the reward system and emotion, and that these neurons therefore may be integral to the pathophysiology of depressive states.

In this study, we employed optogenetic and chemogenetic manipulations, genetic ablation, ex vivo electrophysiology and mouse behavioral assays to address several questions. Specifically, we sought to investigate: 1) the relationship between OT D3 neuron activity and depression-like behaviors, 2) the potential rewarding effect associated with activation of these neurons and elicited self-grooming, and 3) the neural pathway that transmit the activity of OT D3 neuron activity to ultimately influence dopamine release. To answer these questions, we use the chronic restraint stress (CRS) model[47–49] to induce robust depression-like behaviors (increased inactivity and anhedonia) in mice and find that CRS significantly decreases excitability of OT D3 neurons without changing that of neighboring D1/D2-SPNs. Loss-of-function of OT D3 neurons leads to depression-like behaviors (particularly increased inactivity but not anhedonia), whereas activation of these neurons ameliorates CRS-induced depression-like behaviors. Furthermore, optogenetic activation of OT D3 neurons produces conditioned place preference, indicating a rewarding effect, which diminishes when self-grooming is blocked. Finally, we provide ex vivo electrophysiological evidence to support a model in which OT D3 neurons influence dopamine release into the NAc via synaptic connections with OT SPNs that subsequently project to DA neurons in the ventral tegmental area (VTA). Our study reveals a critical role of OT D3 neurons in regulating affective behaviors, suggesting a target for treatment of depression.

## Results
### CRS affects behaviors and reduces OT D3 neuron excitability
To investigate how chronic exposure to stress may influence function of the OT, we subjected double transgenic D1-tdTomato/D2-EGFP mice and transgenic D3-Cre/tdTomato mice to CRS for 14 consecutive days (2 h per day). Twenty-four hours after the last CRS session, the mice underwent multiple behavioral tests (sequentially with stress level from low to high) to assess affective behaviors (Fig. 1a). Since CRS caused similar behavioral changes in the two mouse lines used, we pooled the data for statistical analysis in Fig. 1 (see Supplementary Fig. 1 and Supplementary Fig. 2 for data and statistical analysis separated by each mouse line and sex). For anxiety-like behaviors, we used the open field test (OFT), light-dark box transition test (LDT) and elevated zero maze (EZM) test. In the OFT, CRS did not have significant effects on the total distance traveled nor the time spent in the center zone (Fig. 1b), indicating that CRS did not affect general locomotion. In the LDT, CRS mice exhibited similar behaviors compared to untreated controls measured by the latency for the first entry into the dark area and time spent in the dark area (Fig. 1c). In the EZM test, CRS mice showed an increased latency to the first entry into the open sections while shortened the duration stayed in the open sections compared to the controls (Fig. 1d). For depression-like behaviors, we performed the forced swimming test (FST) and tail suspension test (TST). Compared to the controls, CRS mice spent more time in immobility in both tests (Fig. 1e, f). We then conducted the sucrose preference test to investigate CRS-induced anhedonia, another core behavioral symptom of

depression. To avoid potential influence from prior behavioral tests, we used another cohort of mice to conduct this test (same for the following sucrose preference tests). CRS decreased the sucrose preference index to about 73% of the controls (Fig. 1g), indicating anhedonia. Overall, CRS induced some anxiety-like phenotypes and robust depression-like behaviors in mice, in general agreement with previous reports[50,51]. Since stress affects grooming behavior in rodents[46,52–55], we also quantified the time spent in orofacial grooming and found that CRS mice groomed more than the controls (Fig. 1h; see Discussion).

To examine potential effects of CRS on the electrophysiological properties of OT neurons, we performed whole-cell patch-clamp recordings on D1-tdTomato/D2-EGFP SPNs and D3-Cre/tdTomato neurons (hereafter referred to as D1-, D2-SPNs and D3 neurons, respectively) in acute brain slices from control and CRS mice (Fig. 2a, e). Representative traces of firing patterns of D1-, D2-SPNs and D3 neurons are shown in Fig. 2b and Fig. 2F, respectively. Without altering the input resistance (Fig. 2c, g), CRS significantly lowered the firing frequencies of D3 neurons but not D1- and D2-SPNs upon current injections (Fig. 2d, h), indicating that CRS specifically reduces excitability of D3 neurons in the OT.

### Ablation or inhibition of D3 neurons affects behaviors
We next explored the potential involvement of OT D3 neurons in affective behaviors in physiological conditions (without CRS treatment) by genetically ablating these neurons. We bilaterally injected the Cre-dependent DTA virus or control virus into the OT of D3-Cre/ChR2 mice (ChR2-EYFP as a marker for D3 neurons) (Fig. 3a). We previously showed that four weeks later, this approach efficiently ablated OT D3 neurons revealed by reduced EYFP signals[43]. We performed the same behavioral tests on DTA and control virus mice four weeks post injection (Fig. 3b) and found that ablation of D3 neurons did not affect anxiety-like behaviors (Fig. 3c–e). By contrast, ablation of these neurons induced depression-like behaviors, leading to significantly longer immobility time in both the FST and TST (Fig. 3f, g), similar to CRS effects. Interestingly, ablation of D3 neurons did not change the sucrose preference index (Fig. 3h), suggesting that these neurons are dispensable for this hedonic behavior. As we previously reported[43], ablation of OT D3 neurons significantly reduces total grooming time.

In addition, we tested the potential role of OT D3 neurons in mediating affective behaviors by chemogenetic manipulations. We bilaterally injected a mixture of Cre-dependent excitatory DREADD hM3D(Gq) and inhibitory DREADD KORD (Gi coupled DREADD based on the kappa-opioid receptor template) viruses into the OT of D3-Cre mice (Fig. 4a; Supplementary Fig. 3) to achieve bidirectional manipulations of D3 neuronal activity in the same mice[56] (the results of excitatory DREADD are described later). Ex vivo patch-clamp recordings confirmed that KORD-expressing D3 neurons were inhibited by its ligand salvinorin B (SALB), reflected by significantly decreased firing frequencies compared to the control condition (Fig. 4b). Three weeks after viral injection, we conducted the same behavioral tests with subcutaneous injection of either DMSO (as control) or SALB (Fig. 4a). Although ablation of OT D3 neurons did not result in anxiety-like behaviors in all three tests, inhibition of these neurons led to significant differences in the LDT test, characterized by a shorter latency to the first entry into the dark box and longer time spent in the dark box (Fig. 4d), but no significant differences in the OFT and EZM test (Fig. 4c, e). Similar to ablation of OT D3 neurons, chemogenetic inhibition of these neurons also induced robust depression-like behaviors. SALB injected mice spent longer time in immobility in both the FST and TST compared to DMSO control conditions (Fig. 4f, g). The altered immobility duration was not due to the impaired motor ability since inhibition of OT D3 neurons did not affect general locomotion in the OFT (Fig. 4c). In addition, inhibition of D3 neurons did not affect the sucrose preference index (Fig. 4h), consistent with the ablation experiment (c.f. Fig. 3h). To exclude the potential non-specific effect of

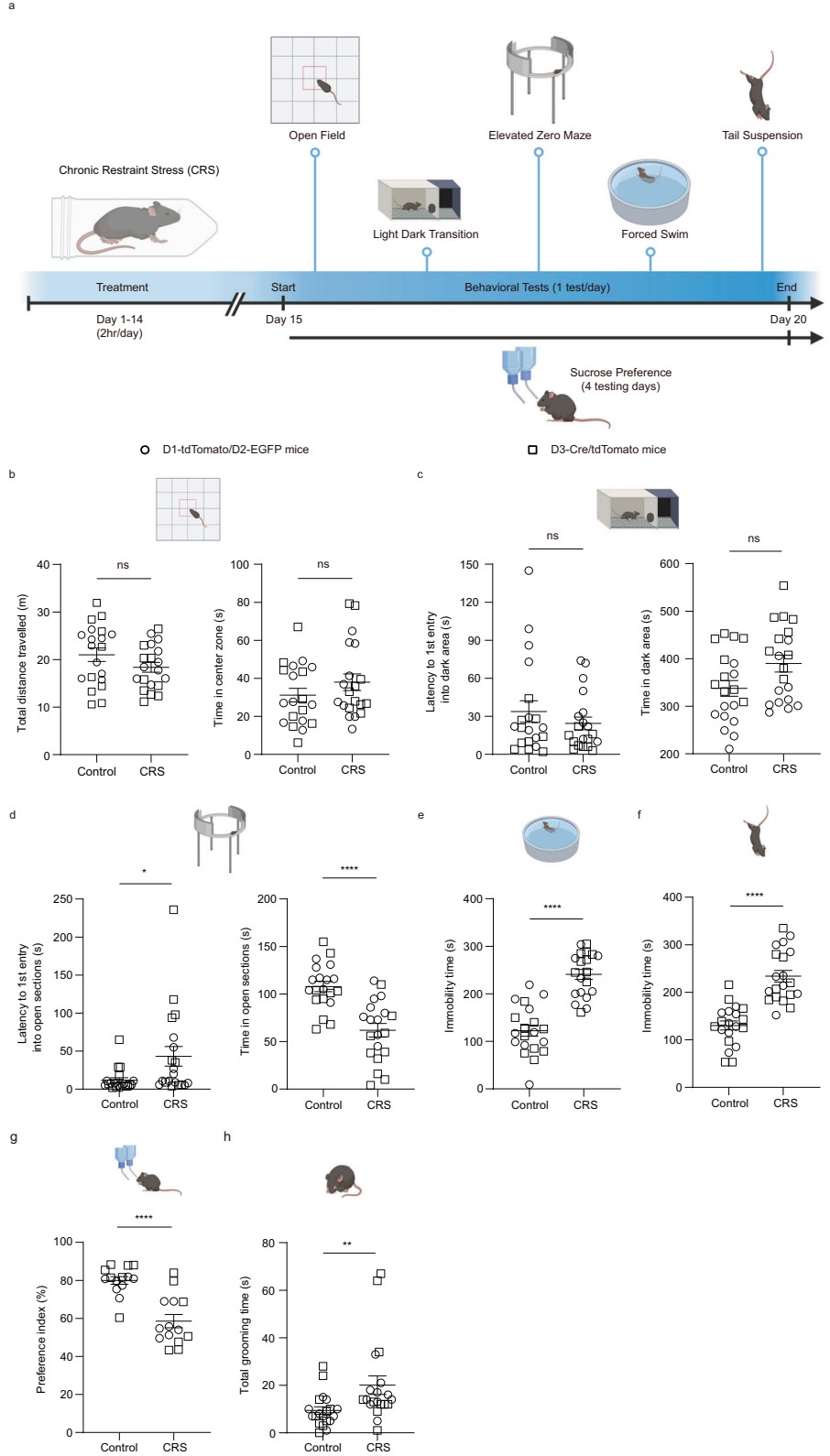

SALB, we bilaterally injected a control virus without DREADD (Cre-dependent AAV8-DIO-mCherry) into the OT of another cohort of D3-Cre mice. Application of DMSO or SALB did not change behaviors in these control mice (Supplementary Fig. 4), supporting that inhibitory DREADD-induced effects result from the interaction between SALB and KORD. Compared to other control conditions, we noticed a reduced total distance traveled with injection of DMSO (Fig. 4 and

Supplementary Fig. 4), presumably due to its toxicity. Since each experimental group has its own DMSO control, the difference between DMSO and SALB should reflect the effect of SALB-mediated action on KORD (Fig. 4). Taken together, these results indicate that loss-of-function of OT D3 neurons via both genetic ablation and chemogenetic inhibition reliably induces depression-like behaviors. Moreover, chemogenetic inhibition of OT D3 neurons decreased grooming behavior

**Fig. 1 | Chronic restraint stress (CRS) changes affective behaviors in mice.**
**a** Experimental strategy and timeline for behavioral assays. Created with BioRender.com. **b–h** Effects of CRS on behavioral performance in different tests. Since CRS caused similar behavioral changes in the two mouse strains used, we pooled the data for statistical analysis (see Supplementary Figs. 1 and 2 for data in each mouse line and each sex). **b** Open field test: total distance traveled (left; $t_{(38)} = 1.507$ and $p = 0.840$) and time in the center zone (right; $t_{(38)} = 1.203$ and $p = 1.000$). **c** Light-dark box transition test: latency to the first entry into the dark area (left; $p = 1.000$) and time spent in the dark area (right; $t_{(38)} = 2.152$ and $p = 0.228$). **d** Elevated zero maze test: latency to the first entry into open sections (left; $p = 0.048$) and time in the open sections (right; $t_{(38)} = 5.129$ and $p = 5.33 \times 10^{-5}$).

**e** Forced swimming test: immobility time ($t_{(38)} = 7.599$ and $p = 3.90 \times 10^{-9}$). **f** Tail suspension test: immobility time ($t_{(38)} = 6.693$ and $p = 6.43 \times 10^{-8}$). **g** Sucrose preference test: preference index ($p = 7.82 \times 10^{-5}$). **h** Total grooming time in the open field test ($p = 2.77 \times 10^{-3}$). $n = 10$ D1-tdTomato/D2-EGFP mice (circle) and 10 D3-Cre/tdTomato mice (square) in **b–f**, **h**, and $n = 7$ per mouse line in **g**. Different cohorts of mice were used in **b–f**, **h** and **g**. Data are expressed as mean ± SEM. Student's two-tailed unpaired $t$ test for **b** and time in **c–f**; two-sided Mann–Whitney test for latency in **c**, **d** and **g**, **h**. $P$ values in **b–d** are adjusted by the Bonferroni correction. *$p < 0.05$, **$p < 0.01$, ****$p < 0.0001$; ns not significant. Source data are provided in the Source Data file.

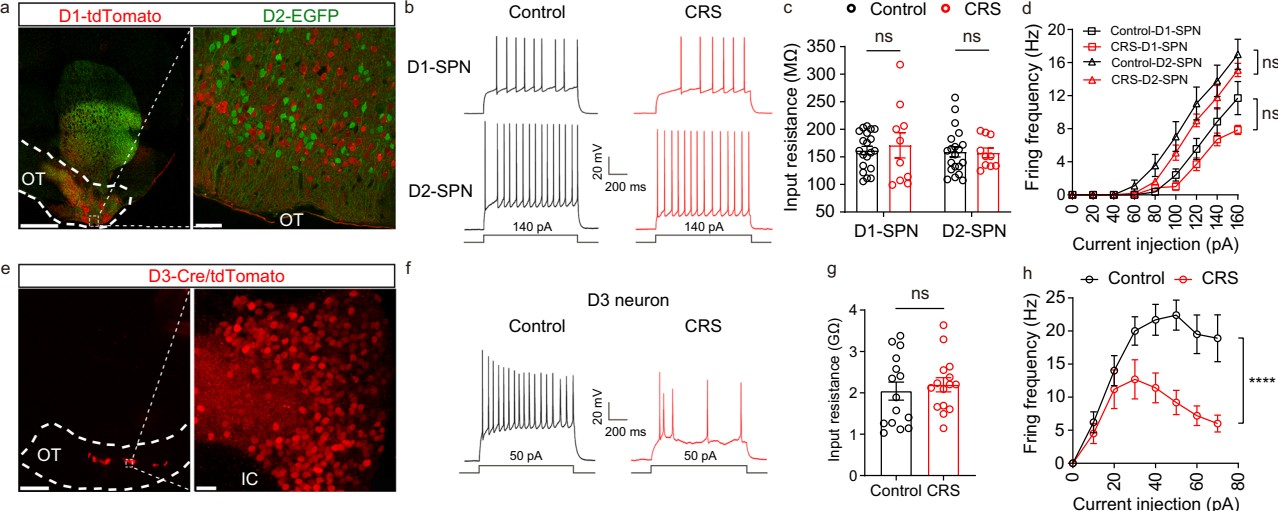

**Fig. 2 | Chronic restraint stress (CRS) significantly decreases neuronal excitability of OT D3 neurons, but not D1- and D2-SPNs.** Confocal images from the olfactory tubercle (OT) showing D1-tdTomato and D2-EGFP SPNs (**a**) and D3-Cre/tdTomato neurons (**e**). Left, low-magnification images. Scale bar: 1 mm. Right, enlarged images. Scale bars: 50 μm (**a**) and 20 μm (**e**). Similar patterns were observed in 3 mice from each line. Patch-clamp recordings showing current injection-induced firing of D1- and D2-SPNs (**b**) and D3 neurons (**f**) from controls (black) and CRS-treated (red) mice. Input resistance of D1-tdTomato and D2-EGFP SPNs (**c**) and D3-tdTomato neurons (**g**). D1-SPNs: $t_{(28)} = 0.486$ and $p = 0.631$. D2-SPNs: $t_{(28)} = 0.127$ and $p = 0.900$. $n = 20$ and 10 for control and CRS, respectively. D3 neurons: $t_{(28)} = 0.552$ and $p = 0.585$; $n = 15$ per group. Firing frequency of D1- and D2-SPNs (**d**) and D3 neurons (**h**) upon current injections from controls and CRS-treated

mice. D1-SPNs: treatment, $F_{(1, 25)} = 0.824$ and $p = 0.373$; current, $F_{(8, 200)} = 72.813$ and $p < 0.1 \times 10^{-13}$; treatment × current, $F_{(8, 200)} = 0.971$ and $p = 0.460$; $n = 20$ and 7 neurons for 5 control and 4 CRS mice, respectively. D2-SPNs: treatment, $F_{(1, 28)} = 0.898$ and $p = 0.351$; current, $F_{(8, 220)} = 26.897$ and $p < 0.1 \times 10^{-13}$; treatment × current, $F_{(8, 220)} = 0.534$ and $p = 0.830$; $n = 20$ and 10 neurons for 5 control and 4 CRS mice, respectively. D3 neurons: treatment, $F_{(1, 18)} = 10.289$ and $p = 0.005$; current, $F_{(3, 48)} = 30.674$ and $p < 0.1 \times 10^{-7}$; treatment × current, $F_{(7, 126)} = 6.451$ and $p = 1.68 \times 10^{-5}$; $n = 10$ neurons each condition from 5 control and 5 CRS mice. Holding membrane potential = −60 mV for all recordings. Data are expressed as mean ± SEM. Student's two-tailed unpaired $t$ tests for **c** and **g**. Two-way ANOVA for **d** and **h**. ****$p < 0.0001$; ns not significant. Source data are provided in the Source Data file.

(Fig. 4i), which potentially influences the behavioral measurements, including overestimation of the immobility time in the FST and TST. We therefore rectified potential grooming-related deviations upon chemogenetic inhibition (detailed in Methods) and similar conclusions could be drawn (Supplementary Fig. 5a-e).

**Activation of D3 neurons normalizes CRS-induced behaviors**
In order to evaluate whether activation of OT D3 neurons can mitigate CRS-induced affective behaviors, we used both optogenetic and chemogenetic approaches. In D3-Cre/ChR2 mice, the islands of Calleja D3 neurons in the OT were visualized by EYFP signals (Fig. 5a, top). These D3-Cre/ChR2 neurons reliably fired action potentials upon blue light stimulation at 20 Hz (473 nm; 10 ms pulse)[43] (Fig. 5a, bottom), and these parameters were applied in the following behavioral tests. Immediately after each CRS session, blue (for activating ChR2-expressing D3 neurons) or green light (less efficiency in activating ChR2 as comparison) was applied for 15 min in the home cage, and all behavioral tests were conducted during blue or green light stimulation, except for the sucrose preference test in which photostimulation was applied immediately before the test (Fig. 5b). Similar to green light condition, activation of D3 neurons by blue light did not alleviate CRS-

induced anxiety-like phenotypes (Fig. 5c–e). By contrast, optogenetic activation of D3 neurons ameliorated CRS-induced depression-like behaviors in both the FST and TST. Compared to green light, activation of D3 neurons by blue light decreased the immobility time by 57% (FST) and 63% (TST) (Fig. 5f, g). Furthermore, optogenetic activation of D3 neurons (15 min/day) was not sufficient to weaken CRS-induced anhedonia in the sucrose preference test (Fig. 5h), consistent with the results from the ablation and inhibition experiments (c.f. Figs. 3h and 4h). Since blue light activation of OT D3 neurons induces grooming[43,57], it potentially influences behavioral measurements, including underestimation of the immobility time in the FST and TST. We rectified potential grooming-related deviations under light stimulation (detailed in Methods) and similar conclusions could be drawn (Supplementary Fig. 6a–e).

In the above experiment (Fig. 5), OT D3 neurons were optogenetically activated after each CRS session as well as during each behavioral test. We next asked whether activation of these neurons only after each CRS session (but not during behavioral tests) was sufficient to reverse CRS-induced depression-like behaviors. This manipulation did not relieve CRS-induced depression-like behaviors (Supplementary Fig. 7a, e, f), nor did it change anxiety-like behaviors (Supplementary

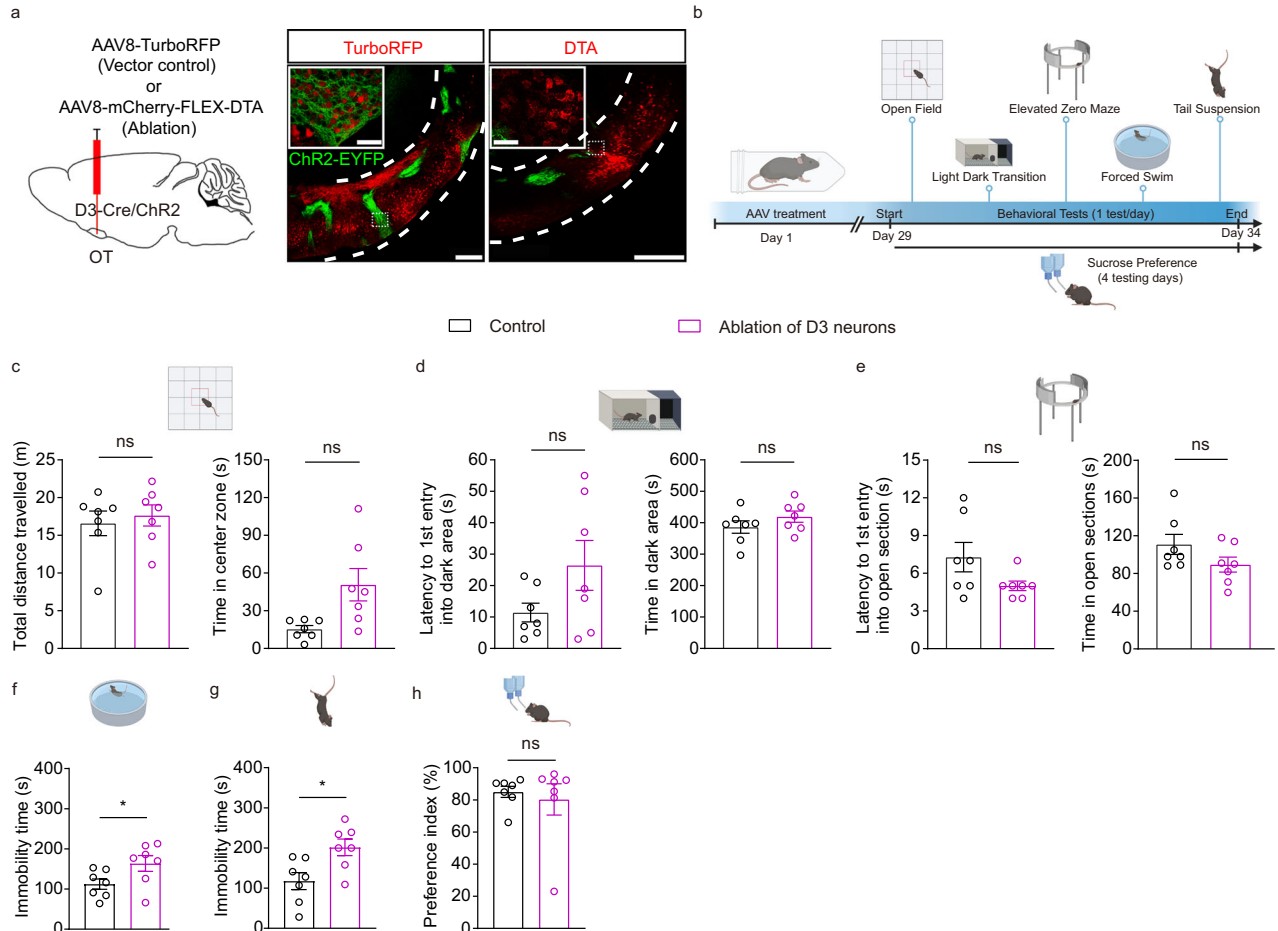

**Fig. 3 | Ablation of OT D3 neurons induces depression-like behaviors. a** Left, schematic showing the viral injection strategy. The AAV8-TurboRFP (control) or Cre-dependent AAV8-mCherry-FLEX-DTA virus (800 nl) was bilaterally injected into the OT. Compared to the control virus (middle), the DTA virus ablated OT D3 neurons as visualized by the absence of most D3-EYFP signals (right). The insets are high magnification images from the corresponding dotted rectangle areas. Confocal images were collected 4 weeks post viral injection. Scale bars = 200 μm (low magnification) and 30 μm (insets). **b** Experimental strategy and timeline of behavioral assays. Created with BioRender.com. **c–h** Effects of ablation of OT D3 neurons on behavioral performance in different tests. **c** Open field test: total distance traveled (left; $t_{(12)} = 0.487$ and $p = 1.000$) and time in the center zone (right;

$t_{(12)} = 2.682$ and $p = 0.120$). **d** Light-dark box transition test: latency to the first entry into the dark area (left; $t_{(12)} = 1.767$ and $p = 0.618$) and time in the dark area (right; $t_{(12)} = 1.244$ and $p = 1.000$). **e** Elevated zero maze test: latency to the first entry into open sections (left; $t_{(12)} = 1.860$ and $p = 0.528$) and time in open sections (right; $t_{(12)} = 1.607$ and $p = 0.804$). **f** Forced swimming test: immobility time ($t_{(12)} = 2.200$ and $p = 0.048$). **g** Tail suspension test: immobility time ($t_{(12)} = 2.854$ and $p = 0.015$). **h** Sucrose preference test: preference index ($t_{(12)} = 0.453$ and $p = 0.658$). Two different cohorts of mice were used in **c–g** and **h**. $n = 7$ mice per group. Data are expressed as mean ± SEM. Student's two-tailed unpaired $t$ tests. $P$ values in **c–e** are adjusted by the Bonferroni correction. *$p < 0.05$; ns not significant. Source data are provided in the Source Data file.

Fig. 7a–d). These findings suggest that optogenetic activation of D3 neurons only after each CRS session is not sufficient to reverse CRS-induced behavioral changes. The relieving effects observed in Fig. 5 are likely due to OT D3 neuron activation during the behavioral tests.

To further support this notion, we next tested the effects of chemogenetic activation of D3 neurons only during the behavioral tests on CRS mice. We performed the same behavioral assays on CRS, excitatory DREADD hM3D(Gq) mice with intraperitoneal injection of either saline (as control) or CNO (activating OT D3 neurons) (Fig. 6a). Ex vivo patch-clamp recordings confirmed that hM3D(Gq)-expressing D3 neurons were activated by its ligand CNO, reflected by significantly increased firing frequencies compared to the control condition (Fig. 6b). Chemogenetic activation of D3 neurons did not mitigate any CRS-induced anxiety phenotypes (Fig. 6c–e). By contrast, activation of these neurons normalized CRS-induced depression-like behaviors, characterized by less immobility time in both the FST and TST (Fig. 6f, g). Similar to optogenetic manipulations, chemogenetic activation of D3 neurons also did not improve CRS-induced anhedonia in the sucrose preference test (Fig. 6h). To exclude the potential non-specific effect of CNO, we

bilaterally injected the Cre-dependent AAV8-DIO-mCherry virus into the OT of another cohort of D3-Cre mice as controls. Intraperitoneal injection of saline or CNO did not influence behaviors of these control mice (Supplementary Fig. 8), supporting that excitatory DREADD-induced effects are ascribed to the interaction between CNO and hM3D(Gq). Moreover, chemogenetic activation of OT D3 neurons increased grooming (Fig. 6i), which may influence behavioral measurements including underestimation of the immobility time in the FST and TST. We therefore rectified potential grooming-related deviations upon chemogenetic activation (detailed in Methods), and similar conclusions could be drawn (Supplementary Fig. 9a–e).

### D3 neuron activation induces conditioned place preference

Since optogenetic or chemogenetic activation of OT D3 neurons has antidepressant effects, we then asked whether activation of these neurons has any inherently rewarding effects (or associated positive valence). We used the conditioned place preference (CPP) assay, a Pavlovian conditioned paradigm, which contained four sessions in four days. In the pre-conditioning session, a mouse was allowed to

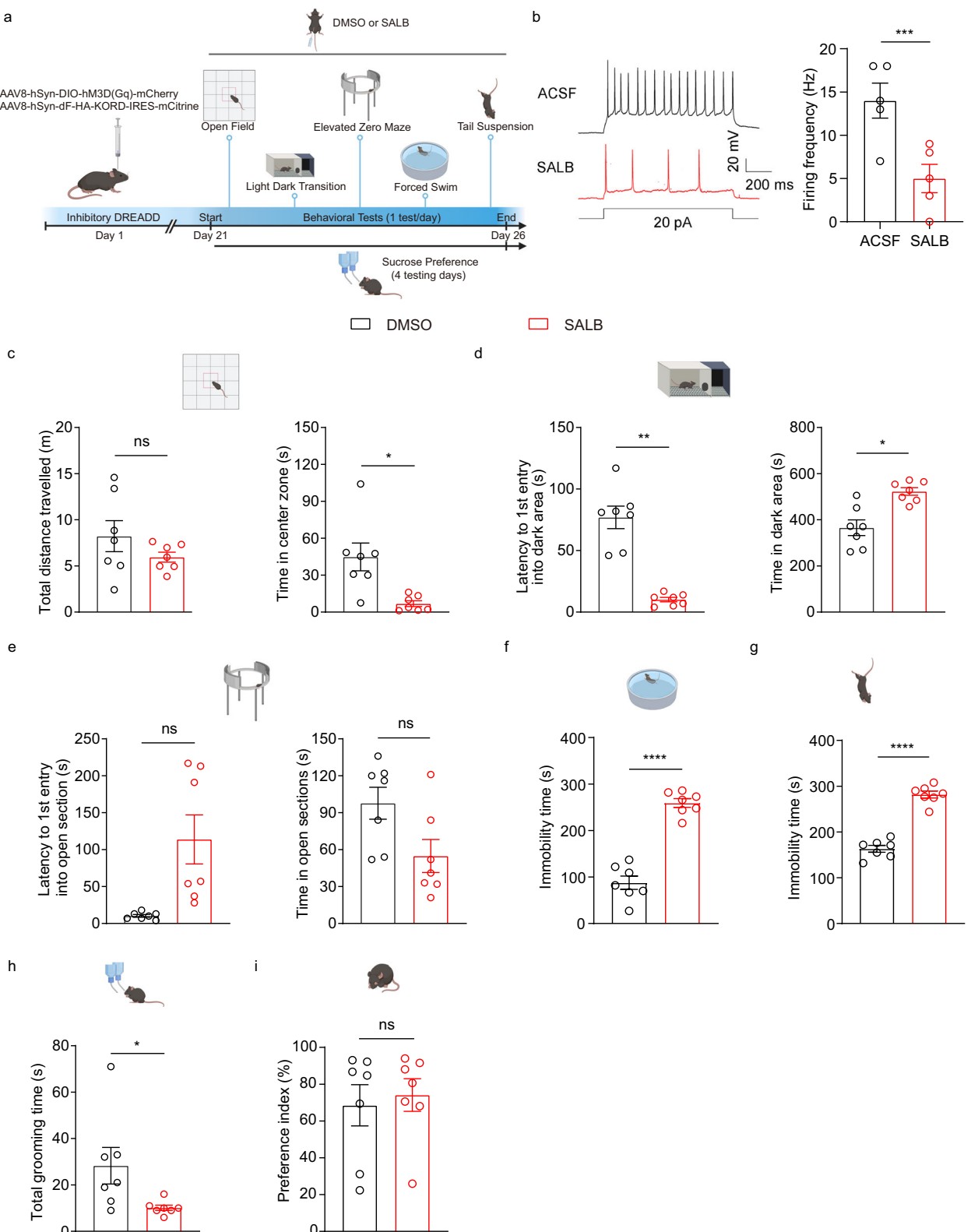

freely explore the three-compartment arena to determine its most and least preferred side chamber (Fig. 7a). In the following two conditioning sessions, the mouse was stimulated by blue light in the least preferred chamber. In the last session, the post-conditioning session, the mouse was again allowed to freely explore the arena, the time it spent in each compartment was measured and the CPP difference score (the time difference between the post- and pre-conditioning session in a chamber) was calculated (Fig. 7b). Unlike the D3-Cre

control mice, which maintained their initial bias for the most preferred chamber, D3-Cre/ChR2 mice spent more time in the laser-paired chamber in the post-conditioning session (Fig. 7c), and thus had a significantly higher CPP score (Fig. 7a, b). We used the commonly adapted CPP test which assigns the least preferred side as the conditioning side to rule out the possibility that post-conditioning preference is due to baseline preference. This design may have the potential limitation due to "regression to the mean", which contributes

**Fig. 4 | Chemogenetic inhibition of OT D3 neurons changes affective behaviors in mice. a** Schematic showing strategy for viral injection into the OT and timeline for behavioral assays under inhibitory DREADD manipulations. Created with BioRender.com. **b** Ex vivo electrophysiological recordings on KORD-expressing D3 neurons. Left, representative traces showing current injection-induced firing under bath application of ACSF or SALB (in DMSO; 10 μM). Right, comparison of the firing frequency between ACSF and SALB condition ($t_{(4)} = 12.728$ and $p = 2.20 \times 10^{-4}$). $n = 5$ neurons from 3 mice. **c–i** Effects of inhibition of D3 neurons on behavioral performance in different tests. **c** Open field test: total distance traveled (left; $t_{(6)} = 1.816$ and $p = 0.714$) and time in the center zone (right; $t_{(6)} = 2.955$ and $p = 0.150$). **d** Light-dark box transition test: latency to the first entry into the dark area (left; $t_{(6)} = 7.773$ and $p = 1.43 \times 10^{-3}$) and time in the dark area (right; $t_{(6)} = 4.641$ and $p = 0.024$).

**e** Elevated zero maze test: latency to the first entry into open sections (left; $t_{(6)} = 3.098$ and $p = 0.126$) and time in open sections (right; $t_{(6)} = 2.511$ and $p = 0.276$). **f** Forced swimming test: immobility time ($t_{(6)} = 11.978$ and $p = 2.05 \times 10^{-5}$). **g** Tail suspension test: immobility time ($t_{(6)} = 10.563$ and $p = 4.23 \times 10^{-5}$). **h** Sucrose preference test: preference index ($t_{(6)} = 0.722$ and $p = 0.497$). **i** Total grooming time in the open field test ($t_{(6)} = 2.579$ and $p = 0.042$). $n = 7$ mice per group. Two different cohorts of mice were used in **c–g**, **i** and **h**. Data are expressed as mean ± SEM. Student's two-tailed paired $t$ tests. $P$ values in **c–e** are adjusted by the Bonferroni correction. *$p < 0.05$, **$p < 0.01$, ***$p < 0.001$, ****$p < 0.0001$; ns not significant, ACSF artificial cerebrospinal fluid solution, DREADD Designer Receptors Exclusively Activated by Designer Drugs, DMSO dimethyl sulfoxide, SALB salvinorin B. Source data are provided in the Source Data file.

to the post-conditioning preference. However, this effect unlikely explains the CPP showed by D3-Cre/ChR2 mice after blue light conditioning since control D3-Cre mice maintained their preference between the post- and pre-conditioning session in the same design (Fig. 7b). Taken together, these results suggest that optogenetic activation of OT D3 neurons has a rewarding effect.

We previously reported that optogenetic activation of OT D3 neurons reliably induced self-grooming while inactivation of these neurons halted ongoing grooming[43]. Similarly, chemogenetic manipulations of these neurons also bidirectionally mediated grooming (Figs. 4i and 6 i). We then asked whether grooming induced by D3 neuron activation is required for the ability of these neurons to drive CPP. We performed the same CPP test using D3-Cre/ChR2 mice with a collar around the neck to block grooming by preventing the forepaws from contacting the face and head (Fig. 7d, inset). Blocking self-directed orofacial grooming eliminated CPP caused by stimulation of OT D3 neurons (Fig. 7d, e), suggesting that grooming elicited by activation of D3 neurons is necessary for the rewarding effect.

### OT D3 neurons affect dopamine release to NAc

We propose a neural circuit model that may explain bidirectional regulation of grooming and depression-like behaviors via the activity of OT D3 neurons (Fig. 8a). OT D3 neurons directly inhibit OT SPNs which in turn inhibit the ventral tegmental area (VTA) dopamine neurons that mediate dopamine release into the NAc. This model is based on several lines of evidence in the literature: 1) self-grooming (both spontaneous and stress-elicited) induces transient dopamine release into the NAc[46], 2) the VTA→NAc pathway is implicated in regulating depression-like behaviors[58–60], 3) OT D3 neurons are local interneurons and provide direct inhibition onto OT D1/D2 SPNs[43], and 4) OT SPNs project directly to VTA[61].

To provide direct evidence to support this model, we examined whether OT SPNs make monosynaptic connections onto NAc-projecting VTA neurons. Since the VTA receives denser innervation from OT D1-SPNs than D2-SPNs[61], we tested functional connections of the OT SPNs→VTA→NAc pathway in D1-Cre mice. We bilaterally injected Cre-dependent AAV1-EF1a-DIO-ChR2-EYFP virus and cholera toxin subunit B-555 (CTB) into the OT and NAc, respectively (Fig. 8b). The ChR2-EYFP$^+$ OT D1-SPNs, CTB$^+$ NAc neurons, and retrogradely labeled CTB$^+$ VTA neurons surrounded with dense D1-SPNs axonal fibers were confirmed post mortem (Fig. 8c, d). We performed whole-cell patch-clamp recordings on CTB$^+$ VTA neurons in acute brain slices and recorded blue light-evoked inhibitory postsynaptic currents (IPSCs, which were inward currents due to high intrapipette [Cl$^-$]). In ~72% (43 out of 60) of CTB$^+$ VTA neurons, repeated light pulses evoked IPSCs, which had short latency (~5 ms) and little jitter (<1 ms) (Fig. 8e). These currents were blocked by GABA$_A$ receptor antagonist bicuculline but not changed by glutamate receptor antagonists, (2R)-amino-5-phosphonovaleric acid (AP5) and cyanquixaline (6-cyano-7-nitroquinoxaline-2,3-dione) (CNQX) (Fig. 8f), supporting the existence of GABA$_A$ mediated monosynaptic connections from OT D1-SPNs onto these NAc-projecting VTA neurons.

Among synaptically connected CTB$^+$ VTA neurons, we classified them into several types based on electrophysiological properties[62,63]. The majority (~67% or 29 out of 43) displayed characteristics of dopamine neurons with low firing frequency (<5 Hz) upon current injection, apparent spike frequency adaptation and voltage "sag". Another 23% were deemed as GABAergic neurons as they had higher firing frequency (≥5 Hz) with little spike frequency adaption or voltage "sag", while 9% belonged to either class (Fig. 8g). Further post-hoc immunostaining with a tyrosine hydroxylase (TH) antibody confirmed that at least some of CTB$^+$ VTA neurons were also TH$^+$ (Fig. 8d), supporting their identity as dopamine neurons. Taken together, these results support that OT SPNs make direct inhibitory synaptic connections onto NAc-projecting VTA neurons.

## Discussion

By using cell-type-specific manipulations, ex vivo electrophysiology and behavioral assays, we reveal that OT D3 neurons (mostly in the islands of Calleja) bidirectionally mediate depression-like behaviors in mice. Loss-of-function of OT D3 neurons leads to depression-like behaviors (specifically increased inactivity), gain-of-function of these neurons alleviates CRS-induced changes in affective behaviors. We propose a model which links the activity of OT D3 neurons and self-grooming with the reward system and stress-induced responses.

We first examined the effects of CRS treatment using multiple behavioral assays to assess affective behaviors in mice. CRS induced robust depressive phenotypes, (i.e., increased immobility in both forced swimming and tail suspension tests) and anhedonia (decreased preference for sucrose) (Fig. 1e–g), supporting that CRS effectively causes depression-like behaviors in rodents as previously reported[47–49,64]. In addition to depressive phenotypes, CRS mice also exhibited anxiety-like behaviors. Unlike the consistent performance in depression-like behavior assays (FST and TST), CRS mice exhibited divergent states in different anxiety assays. Anxiety-like behaviors were observed in the EZM test, but not in the OFT and LDT test (Fig. 1b–d), suggesting that these behavioral assays may have different sensitivities or test different aspects of anxiety. Given that there is no single ideal mouse test for anxiety and that each existing test has its advantages[65], a combination of different behavioral tests produces a better understanding in anxiety-related processes[66]. Our findings further support the necessity of using multiple behavioral assays for testing anxiety-like behaviors.

CRS specifically decreases excitability of OT D3 neurons but not OT D1- and D2-SPNs (Fig. 2), which is in sharp contrast to results from NAc circuits. Stress produces distinct changes (e.g., morphology, excitability, synaptic transmission) in NAc D1- and D2-SPNs, which have opposing roles in depression-like behaviors[2]. Specifically, chronic stress induces hyperexcitability of NAc D1-SPNs[20,21]. Here we report that OT D3 neurons are vulnerable but OT SPNs are resilient to stressful and depressive states, suggesting chronic stress exerts differential influences on the OT and NAc circuitry. The role of dopamine receptor-expressing neurons in depression-like behaviors is also implicated in other brain areas. For example, dysfunctions of p11 in dopamine D2

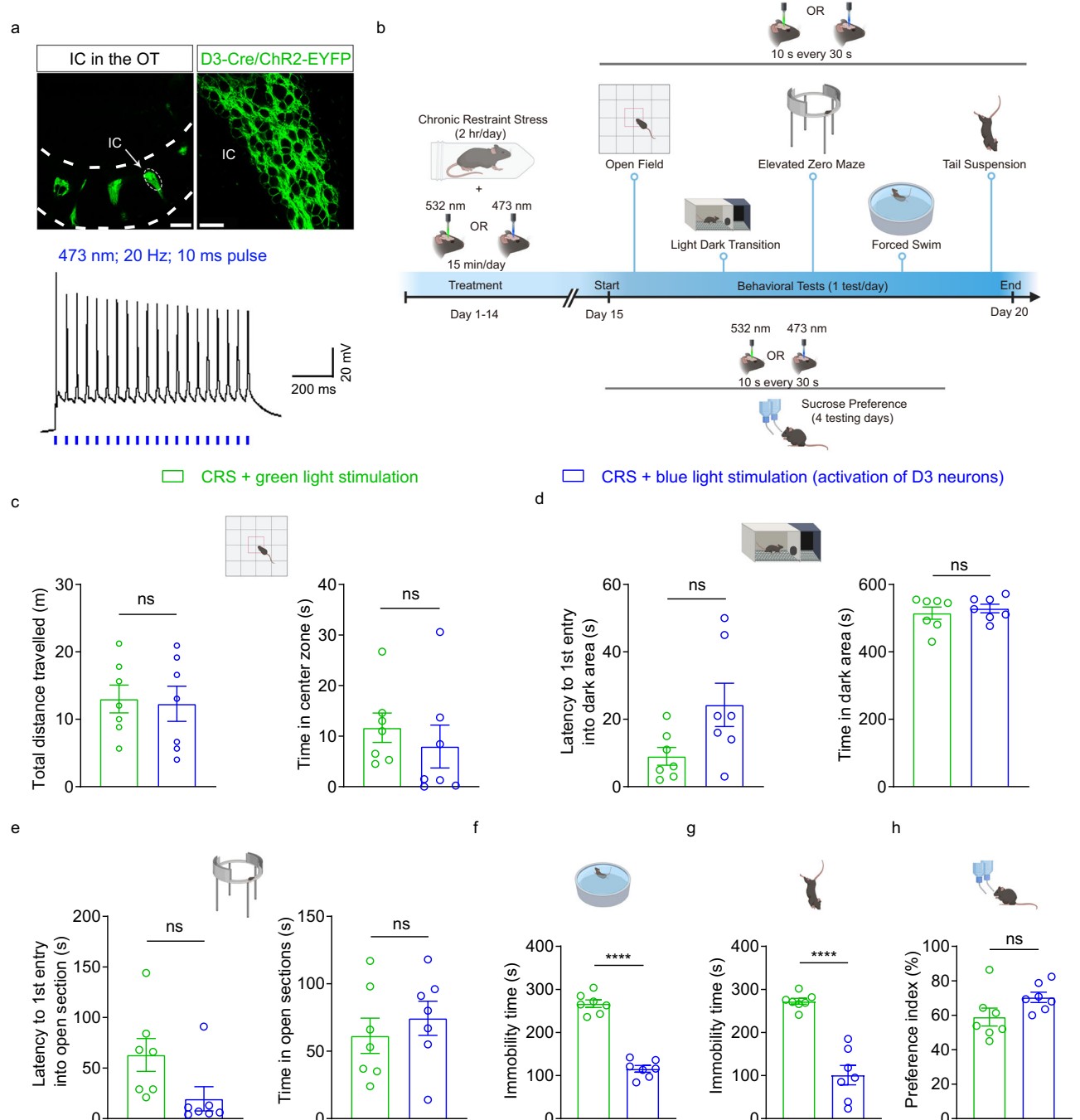

**Fig. 5 | Optogenetic activation of OT D3 neurons normalizes CRS-induced depression-like behaviors. a** D3 neurons reliably fire action potentials upon blue light stimulation. Top: individual islands of Calleja (IC) in the OT (left) and enlarged view of the dashed area showing tightly packed D3-Cre/ChR2-EYFP neurons in IC (right). Scale bars: 200 μm (left) and 20 μm (right). Bottom: a D3-Cre/ChR2 neuron fires reliably upon blue light stimulation at 20 Hz (473 nm; 10 ms pulse). **b** Experimental strategy and timeline of behavioral assays. Created with BioRender.com. **c**–**h** Effects of blue light activation of D3 neurons on behavioral performance of CRS mice in different tests compared to green light stimulation. **c** Open field test: total distance traveled (left; $t_{(12)} = 0.220$ and $p = 1.000$) and time in the center zone (right; two-sided Mann–Whitney test; $p = 1.000$). **d** Light-dark box transition test: latency to the first entry into the dark area (left; $t_{(12)} = 2.197$ and $p = 0.180$) and time in the dark area (right; $t_{(12)} = 0.616$ and $p = 1.000$). **e** Elevated zero maze test: latency to the first entry into open sections (left; two-sided Mann–Whitney test; $p = 0.096$) and time in open sections (right; $t_{(12)} = 0.713$ and $p = 1.000$). **f** Forced swimming test: immobility time ($t_{(12)} = 12.908$ and $p = 2.14 \times 10^{-8}$). **g** Tail suspension test: immobility time ($t_{(12)} = 7.278$ and $p = 9.78 \times 10^{-6}$). **h** Sucrose preference test: preference index ($t_{(12)} = 1.904$ and $p = 0.081$). $n = 7$ mice per group. Two different cohorts of mice were used in **c**–**g** and **h**. Data are expressed as mean ± SEM. Student's two-tailed unpaired $t$ tests. $P$ values in **c**–**e** are adjusted by the Bonferroni correction. ****$p < 0.0001$; ns not significant. Source data are provided in the Source Data file.

receptor-expressing neurons in the lateral habenula and prelimbic cortex contribute to depression-like behaviors[67,68]. Our results suggest a causal relationship between activity of OT D3 neurons and depressive phenotypes, which is supported by both loss-of-function and gain-of-

function experiments. Ablation or chemogenetic inhibition of OT D3 neurons caused depressive behaviors (Figs. 3 and 4), especially increased inactivity, whereas optogenetic or chemogenetic activation of these neurons alleviates CRS-induced depression-like behaviors or

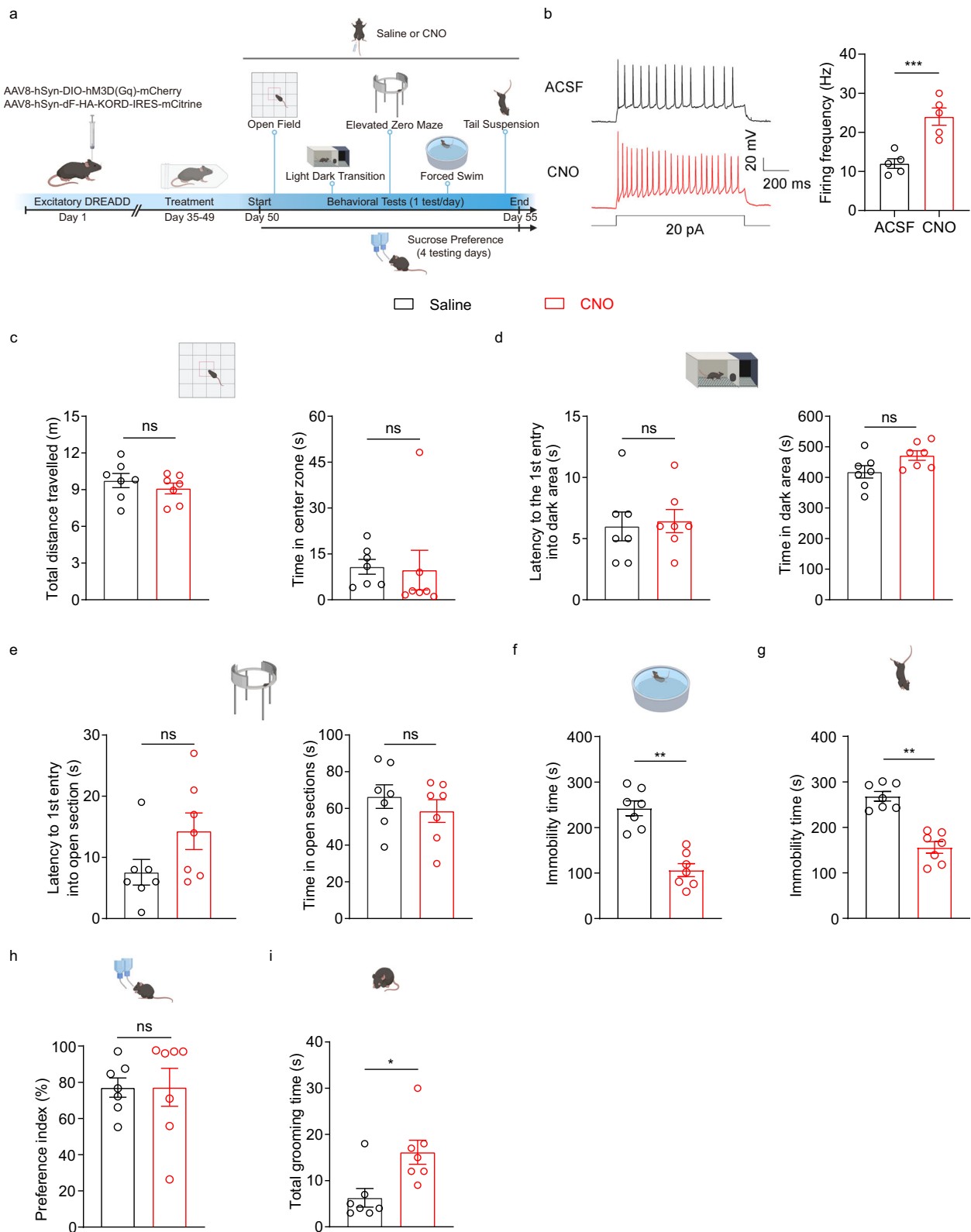

makes the mice more resilient (Figs. 5 and 6). These findings add OT to the existing brain areas that are causally linked to depression-like behaviors. Notably, olfactory bulbectomy in rodents lead to depression-like behaviors[69], but the underlying mechanisms are not fully understood. Since the OT receives direct inputs from the OB, it would be interesting to determine whether OT circuitry contributes to depression-like behaviors in bulbectomized rodents.

In contrast to the striking effects on the immobility time in forced swimming and tail suspension tests, optogenetic or chemogenectic activation of OT D3 neurons is insufficient to ameliorate anhedonia (reduced sucrose preference) in CRS mice (Figs. 5 and 6). One possibility is that different aspects of CRS-induced changes in affective behaviors are mediated by distinct neuronal types and/or brain regions, which is supported by the finding that distinct ventral pallidal

**Fig. 6 | Chemogenetic activation of OT D3 neurons normalizes CRS-induced depression-like behaviors. a** Schematic showing strategy for viral injection into the OT and timeline for behavioral assays under excitatory DREADD manipulations. Created with BioRender.com. **b** Patch clamp recordings on hM3D(Gq)-expressing D3 neurons. Left, representative traces showing current injection-induced firing under bath application of ACSF or CNO (10 μM). Right, comparison of the firing frequency between ACSF and CNO condition ($t_{(4)} = 11.442$ and $p = 3.33 \times 10^{-4}$). $n = 5$ neurons from 3 mice. **c–i** Effects of chemogenetic activation of D3 neurons on behavioral performance in different tests. **c** Open field test: total distance traveled (left; $t_{(6)} = 0.816$ and $p = 1.000$) and time in the center zone (right; $p = 1.000$). **d** Light-dark box transition test: latency to the first entry into the dark area (left; $p = 1.000$) and time in the dark area (right; $t_{(6)} = 1.815$ and $p = 0.720$). **e** Elevated zero maze test: latency to the first entry into open sections (left; $t_{(6)} = 1.943$ and

$p = 0.600$) and time in open sections (right; $t_{(6)} = 1.113$ and $p = 1.000$). **f** Forced swimming test: immobility time ($t_{(6)} = 5.739$ and $p = 0.0012$). **g** Tail suspension test: immobility time ($t_{(6)} = 5.623$ and $p = 0.0014$). **h** Sucrose preference test: preference index ($t_{(6)} = 0.722$ and $p = 0.497$). **i** Total grooming time in the open field test ($p = 0.016$). $n = 7$ mice per group. Two different cohorts of mice were used in **c–g**, **i** and **h**. Student's two-tailed paired $t$ tests for **b**, total distance in **c**, time in **d**, and **e–h**. Two-sided Wilcoxon matched-pairs signed rank test for time in **c**, latency in **d**, and **i**. Data are expressed as mean ± SEM. $P$ values in **c–e** are adjusted by the Bonferroni correction. *$p < 0.05$, **$p < 0.01$, ***$p < 0.001$; ns not significant. ACSF artificial cerebrospinal fluid solution, DREADD Designer Receptors Exclusively Activated by Designer Drugs; CNO Clozapine N-oxide. Source data are provided in the Source Data file.

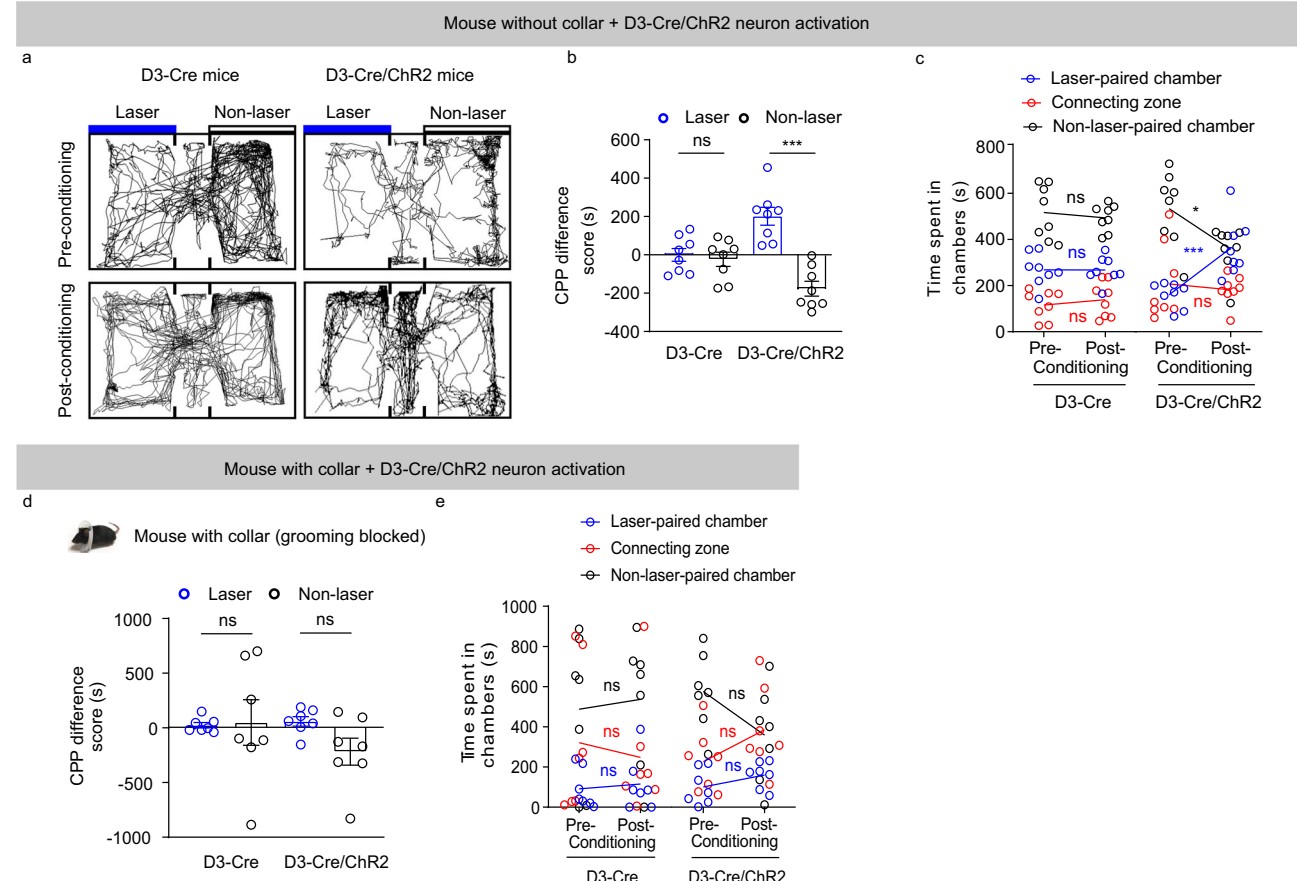

**Fig. 7 | Optogenetic activation of OT D3 neurons induces conditioned place preference (CPP). a** Behavioral tracks of D3-Cre (control) and D3-Cre/ChR2 mice during the pre-conditioning (day 1) and post-conditioning (day 4) sessions. Blue laser stimulation (day 2 and 3) was paired with the least preferred chamber determined in the pre-conditioning session. **b** The CPP difference score. D3-Cre control mice showed no significant difference between the laser-paired and non-laser-paired chamber ($t_{(7)} = 0.470$ and $p = 0.653$). D3-Cre/ChR2 mice had a significantly higher CPP score in the laser-paired than the non-laser-paired chamber ($t_{(7)} = 10.031$ and $p = 2.10 \times 10^{-5}$). **c** Time spent in each compartment. For D3-Cre control mice, no difference between pre- and post-conditioning sessions (laser-paired: $t_{(7)} = 0.472$ and $p = 0.651$; connecting: $t_{(7)} = 0.527$ and $p = 0.614$; non-laser-paired: $t_{(7)} = 0.714$ and $p = 0.499$). D3-Cre/ChR2 mice spent more time in the laser-paired chamber and less time in the non-laser-paired chamber side without change in the connecting zone after conditioning (laser-paired: $t_{(7)} = 5.533$ and

$p = 8.75 \times 10^{-4}$; connecting: $t_{(7)} = 0.407$ and $p = 0.696$; non-laser-paired: $t_{(7)} = 2.980$ and $p = 0.021$). **d** The CPP difference score in the laser-paired or non-laser-paired chamber from mice wearing collar (grooming blocked). Both mouse lines showed no significant difference (D3-Cre: $t_{(6)} = 0.120$ and $p = 0.908$; D3-Cre/ChR2: $t_{(6)} = 1.781$ and $p = 0.125$). **e** Time spent in each compartment from mice wearing collar (grooming blocked). No significant difference between pre- and post-conditioning session. D3-Cre mice: laser-paired: $t_{(6)} = 0.384$ and $p = 0.706$; connecting: $t_{(6)} = 0.413$ and $p = 0.687$; non-laser paired: $t_{(6)} = 0.270$ and $p = 0.792$. D3-Cre/ChR2 mice: laser paired: $t_{(6)} = 1.422$ and $p = 0.181$; connecting: $t_{(6)} = 1.595$ and $p = 0.137$; non-laser paired: $t_{(6)} = 1.898$ and $p = 0.082$. $n = 8$ mice per group in **b** and **c**, $n = 7$ mice per group in **d** and **e**. Data are expressed as mean ± SEM. Student's two-tailed paired $t$ tests. *$p < 0.05$, ***$p < 0.001$; ns, not significant. Source data are provided in the Source Data file.

neurons mediate separate depression-like behaviors[70]. Activation of OT D3 neurons may specifically improve "the decreased motivation for activity" in CRS mice, while anhedonic phenotypes might be mediated by other neuronal subtypes such as cholinergic neurons[25,71] and/or

other brain regions. There is evidence supporting that NAc D1-SPNs are critical for the expression of anhedonia[2,26,27]. The other possibility is that OT D3 neurons are involved in mediating CRS-induced anhedonia but our relatively short activation of OT D3 neurons (15 min/day in

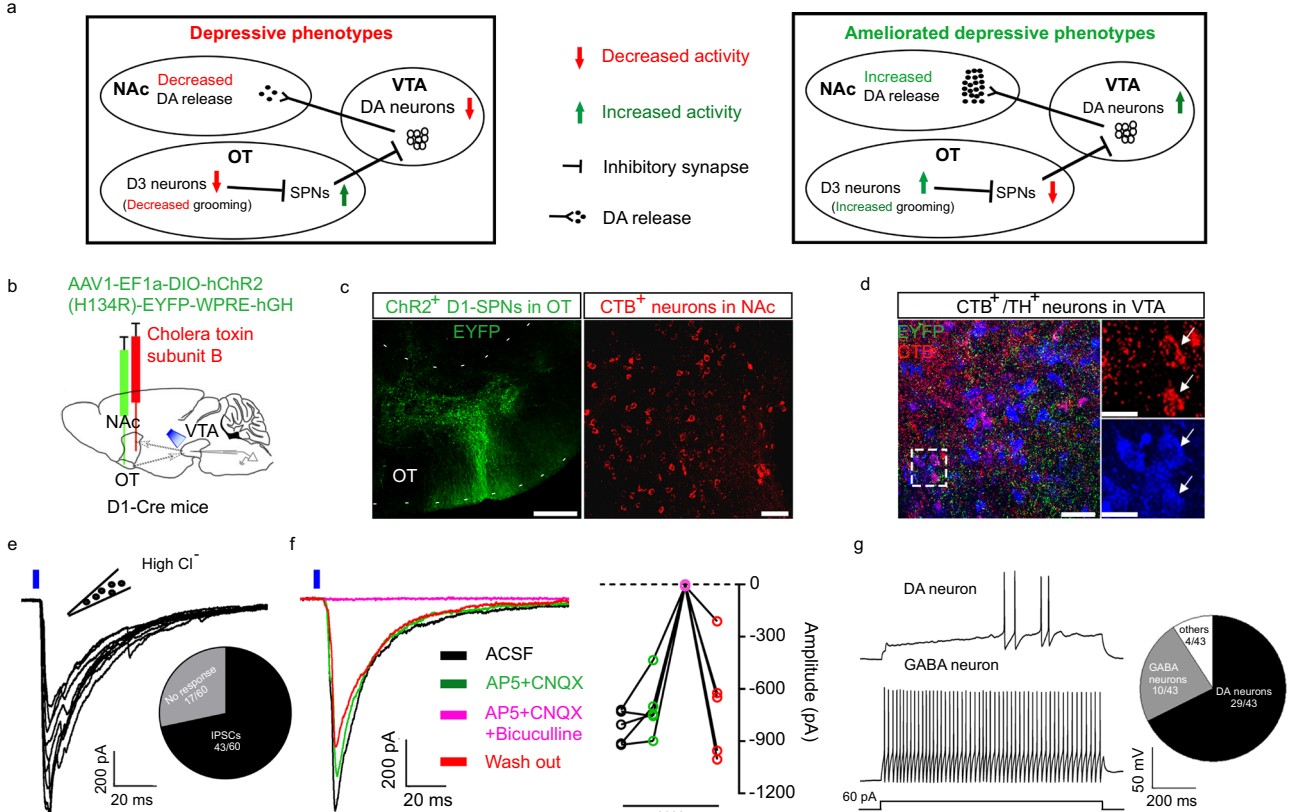

**Fig. 8 | OT SPNs make direct inhibitory synaptic connections with NAc-projecting VTA dopamine neurons. a** A model on how OT D3 neurons bidirectionally mediate affective behaviors in mice. Left, ablation or inhibition of OT D3 neurons disinhibits OT SPNs which in turn inhibit VTA dopamine neurons, reducing dopamine release to NAc. Right, activation of OT D3 neurons inhibits OT SPNs which in turn disinhibit VTA dopamine neurons, enhancing dopamine release to NAc. **b** Strategy for viral injection and retrograde tracing. AAV1 carrying ChR2-EYFP and cholera toxin subunit B were bilaterally injected into the OT and NAc, respectively, in D1-Cre mice ($n = 5$). **c** Confocal images showing ChR2-EYFP[+] OT D1-SPNs (left) and CTB[+] NAc neurons (right). Scale bar = 200 μm (left) and 50 μm (right). **d** Confocal image showing CTB[+] and tyrosine hydroxylase (TH)[+] neurons surrounded with ChR2-EYFP[+] D1-SPN axonal fibers in VTA (left). High magnification images (right) showing two CTB and TH double positive neurons from the dashed rectangle in the left panel. Scale bar = 50 μm (left) and 20 μm (right). **e** D1-SPNs

inhibit VTA neurons. Repeated 10 ms blue laser pulses (0.1 Hz) evoked IPSCs in 71.7% (inset) of VTA neurons recorded. **f** Light evoked IPSCs in VTA neurons were not changed by glutamate receptor antagonists (50 μM AP5 and 20 μM CNQX) but blocked by GABA$_A$ receptor antagonist bicuculline (10 μM). Left, representative traces. Right, quantification of IPSC amplitudes. One-way ANOVA; $F_{(3,16)} = 18.517$ and $p = 2.65 \times 10^{-5}$ (ACSF vs. AP5 + CNQX + bicuculline: $p = 7.35 \times 10^{-6}$; ACSF vs. AP5 + CNQX: $p = 0.378$; ACSF vs. wash out: $p = 0.371$; AP5 + CNQX vs. AP5 + CNQX + bicuculline: $p = 3.69 \times 10^{-5}$; AP5 + CNQX vs. wash out: $p = 0.948$; AP5 + CNQX + bicuculline vs. wash out: $p = 7.72 \times 10^{-5}$). $n = 5$ neurons. ****$p < 0.0001$. **g** Left, representative traces of dopamine (DA) and GABA neurons. Right, pie chart showing the composition of recorded VTA neurons. Baseline membrane potential was kept at −60 mV. Data are expressed as mean ± SEM. Source data are provided in the Source Data file.

optogenetic manipulation in Fig. 5 and approximately 2 h chemogenetic manipulation in Fig. 6) is not sufficient to improve the reduced sucrose preference. This hypothesis is, to some extent, supported by our finding that activation of OT D3 neurons had a rewarding effect in a Pavlovian conditioned paradigm (Fig. 7). A recent report suggests that acute and chronic optogenetic stimulations on certain neural pathways could compensate each other in their effects on normalizing depression-like behaviors[72]. It would be interesting to separate the effects of chronic and repetitive optogenetic activation of OT D3 neurons during CRS treatment versus acute optogenetic activation during the affective behavior assays.

In this study, the two loss-of-function approaches produced somewhat different effects on anxiety-like behaviors. Chemogenetic inhibition of OT D3 neurons induced anxiety-like behaviors in the LDT test but not in the other two tests (Fig. 4). However, ablation of these neurons did not produce anxiogenic effect in all three tests (Fig. 3). Unlike transient inhibition via chemogenetic manipulations, ablation of OT D3 neurons permanently alters the OT circuitry. For instance, it may induce compensatory changes in the neuronal activity of neighboring OT D1- and/or D2-SPNs, which in turn affects both local and

long-range circuits (e.g., the VTA→lateral septum pathway) that are involved in regulating anxiety-related behaviors[73]. Curiously, the anxiety-like behaviors induced by CRS (only in the EZM test) and chemogenetic inhibition of OT D3 neurons (only in the LDT test) are different, suggesting that CRS-induced anxiety-like behaviors engage neural circuits in addition to OT D3 neurons.

Consistent with our previous finding that optogenetic activation or inactivation of OT D3 neurons initiates or halts self-grooming[43], here we demonstrate that chemogenetic manipulations also bidirectionally mediate the total grooming time (Figs. 4 and 6). Interestingly, CRS treatment decreased excitability of OT D3 neurons (Fig. 2), which should act to reduce the grooming drive from these neurons. However, CRS mice exhibited more grooming than controls (Fig. 1h), suggesting that other grooming centers may be more active after CRS treatment to ensure this adaptive behavior in stressed situation[44,46,52,55].

The potential role of D3 receptor in grooming and depression-like behaviors remains elusive. D3 receptor knockout mice display high basal level of grooming[74], consistent with the inhibitory action of this receptor in OT D3 neurons. Whereas some studies suggest that mice lacking D3 receptor are more resistant to stressful conditions and

display normal emotional behaviors[75,76], others show that D3 receptor deficiency results in depression-like behaviors[77,78]. Since D3 receptor is expressed in multiple brain regions, specific manipulation of D3 receptor expression in distinct subpopulations of D3 neurons would be required to dissect out its role.

As OT D3 neurons are local GABAergic interneurons[43], we propose a model which links these neurons and self-grooming with the NAc reward system through OT SPNs and VTA dopamine neurons (Fig. 8). Decreased OT D3 neuron activity would disinhibit monosynaptically connected SPNs, leading to enhanced inhibition onto NAc-projecting VTA dopamine neurons. This would attenuate dopamine release into the NAc and ultimately induce depressive phenotypes in these mice. On the other hand, activation of OT D3 neurons eventually leads to more dopamine release into the NAc, manifested as increased motivation for activity in CRS mice. This model is supported by the rewarding effect of activation of OT D3 neurons, and interestingly, this effect diminishes when grooming is physically blocked (Fig. 7). Furthermore, we provide ex vivo electrophysiological evidence to support that OT D1-SPNs directly inhibit NAc-projecting VTA dopamine neurons (Fig. 8). However, we do not rule out other potential pathways that link OT D3 neurons and grooming with NAc dopamine signaling.

Taken together, we discovered a crucial role of OT D3 neurons in bidirectionally mediating depression-like behaviors. The findings that activation of OT D3 neurons has a rewarding effect and efficiently alleviates CRS-induced depressive phenotypes by increasing the motivation for activity suggest that ventral striatal OT/islands of Calleja D3 neurons are an attractive target for the intervention and treatment of depression.

## Methods

### Animals
The D1-tdTomato[79] and D2-EGFP (Tg(Drd2-EGFP)S118Gsat)[80] mice were crossed to obtain double-transgenic mice in which dopamine D1 and D2 receptor-expressing neurons are labeled with red and green fluorescence, respectively. Bacterial artificial chromosome (BAC) transgenic D3-Cre (Tg(Drd3-cre)KI198Gsat) was obtained from Mutant Mouse Resource & Research Centers (MMRRC) and crossed with the Cre-dependent tdTomato reporter line (JAX Stock No: 007909 or Ai9 line: Rosa26-floxed-tdTomato) or Cre-dependent channel rhodopsin 2 (ChR2) line (JAX Stock No: 024109 or Ai32 line: Rosa26-floxed-ChR2)[81] to generate D3-Cre/tdTomato or D3-Cre/ChR2 mice, respectively. This D3-Cre line was chosen because Cre expression is more restricted to the islands of Calleja neurons[43]. Drd1-Cre (D1-Cre for simplicity; MMRRC Strain# 37156-Jax) mice were obtained from MMRRC[82]. Both male and female mice (8–12 weeks old) were used in all experiments. Mice were housed on a 12 h light/dark cycle with food and water available *ad libitum*. The temperature was maintained between 20° to 26° and the humidity between 40% to 70%. Mice were group-housed until the surgery of viral injection and/or intra-cranial optical fiber implantation and singly-housed afterwards. All animal procedures were in compliance with the governmental regulations in China and USA and approved by the Institutional Animal Care and Use Committee of the Institute of Zoology, Chinese Academy of Sciences, of the University of Pennsylvania, and of the University of Florida.

### Chronic restraint stress model
Chronic restraint stress (CRS) treatment was performed as previously documented[48,67,68]. Briefly, mice were individually placed head first into a well-ventilated 50 ml polypropylene conical tube, which was then plugged with a 4.5-cm-long middle tube, and finally tied with the cap of the 50 ml tube. The restraint stress lasted for 2 h per day (approximately from 9 to 11 am) for consecutive 14 days. After each session of the restraint stress, mice were returned to their home cage, where they were housed in pairs with food and water available *ad libitum*. Twenty-four hours after the last restraint session, mice were subjected to behavioral assays or electrophysiological recordings.

### Virus/toxin injection and optical fiber implantation
Mice were anesthetized with isoflurane (~3% in oxygen) and secured in a stereotaxic system (Model 940, David Kopf Instruments). Isoflurane levels were maintained at 1.5–2% for the remainder of the surgery. Body temperature was maintained at 37 °C with a heating pad connected to a temperature control system (TC-1000, CWE Inc.). Local anesthetic (bupivacaine, 2 mg/kg, s.c.) was applied before skin incision and hole drilling on the dorsal skull. To target the islands of Calleja in the OT, we used two sets of coordinates from bregma: anteroposterior (AP) 1.2 (or 1.54) mm; mediolateral (ML), ±1.1 (or ±1.15) mm; dorsoventral (DV), −5.5 (or −5.0) mm. For the NAc, in order to retrogradely label more NAc-projecting VTA neurons, we injected CTB (recombinant, Alexa Fluor™ 555 conjugate, Invitrogen #C34776) into both the core and shell subregions: core, AP, 1.6 mm; ML, ±0.8 mm; DV, −4.1 mm; medial shell, AP, 1.5 mm; ML, ±0.55 mm; DV, −4.7 mm; lateral shell, AP, 0.98 mm; ML, ±1.8 mm; DV, −4.92 mm. For viral injection, as appropriate, AAV8-CMV-TurboRFP-WPRE-rBG (2.94 × 10^10 vg/ml), AAV8-EF1α-mCherry-FlEX-DTA (3.3 × 10^9 viral units/ml) (University of North Carolina Viral Vector Core, Chapel Hill, NC) (800 nl each side), AAV1-DIO-ChR2-EYFP (C-34777, Life Technologies) (300–500 nl each side), AAV8-hSyn-DIO-hM3D(Gq)-mCherry (≥4 × 10^12 vg/ml; Addgene, cat.# 44361), AAV8-hSyn-dF-HA-KORD-IRES-mCitrine (≥7 × 10^12 vg/ml; Addgene, cat.# 65417) or AAV8-hSyn-DIO-mCherry (≥1 × 10^13 vg/ml; Addgene, cat.# 50459) was bilaterally (300–500 nl each side) injected into the OT via a Hamilton syringe (5 μl) with a flow rate of 50 nl/min controlled by an Ultra Micro Pump III (UMP3) with a SYS-micro4 controller attachment (World Precision, Sarasota, USA). In D1-Cre mice, cholera toxin subunit B (300–500 nl) was also bilaterally injected into NAc. The tip of the syringe was left for 10–15 min after the injection. For optical fiber implantation, a cannula (CFMC14L10-Fiber Optic Cannula, Ø2.5 mm Ceramic Ferrule, Ø400 μm Core, 0.39 NA; Thorlabs, Newton, NJ), customized to 6 mm length, was unilaterally placed in the OT at the coordinates aforementioned and fixed on the skull with dental cement in D3-Cre/ChR2 mice. Mice were returned to home cage for recovery for one week before behavioral tests and mice with viral injection had at least three week waiting period before tests. All optical fiber implantation locations were post-hoc verified and only mice with the intended targeted site were included in data analysis.

### Ex vivo electrophysiological recording
Mice were deeply anesthetized (ketamine-xylazine; 200 and 20 mg/kg body weight, respectively) and quickly decapitated. The dissected brain was immediately placed in ice-cold cutting solution containing (in mM) 92 N-Methyl D-glucamine, 2.5 KCl, 1.2 NaH$_2$PO$_4$, 30 NaHCO$_3$, 20 HEPES, 25 glucose, 5 Sodium L-ascorbate, 2 Thiourea, 3 Sodium Pyruvate, 10 MgSO$_4$, and 0.5 CaCl$_2$; osmolality ~300 mOsm and pH -7.3, bubbled with 95% O$_2$–5% CO$_2$. Coronal sections (250 μm thick) containing the OT were cut using a Leica VT 1200 S vibratome. Brain slices were incubated in oxygenated artificial cerebrospinal fluid (ACSF in mM: 124 NaCl, 3 KCl, 1.3 MgSO$_4$, 2 CaCl$_2$, 26 NaHCO$_3$, 1.25 NaH$_2$PO$_4$, 5.5 glucose, and 4.47 sucrose; osmolality ~305 mOsm and pH -7.3, bubbled with 95% O$_2$–5% CO$_2$) for ~30 min at 31 °C and at least 30 min at room temperature before use. For recordings, slices were transferred to a recording chamber and continuously perfused with oxygenated ACSF. Fluorescent cells were visualized through a 40X water-immersion objective on an Olympus BX61WI upright microscope equipped with epifluorescence. Whole-cell patch-clamp recordings were controlled by an EPC-10 amplifier combined with Pulse v8.74 (HEKA Electronik) and analyzed with Igor Pro 6. Recording pipettes were made from borosilicate glass with a Flaming-Brown puller (P-97, Sutter Instruments; tip resistance 5–10 MΩ). The pipette solution contained (in mM) 120 K-gluconate, 10 NaCl, 1 CaCl$_2$, 10 EGTA, 10

HEPES, 5 Mg-ATP, 0.5 Na-GTP, and 10 phosphocreatine. Light stimulation was delivered through the same objective via pulses of blue laser (473 nm, FTEC2473-V65YF0, Blue Sky Research, Milpitas, USA) with 10 ms light pulse at 20 Hz. For light-evoked inhibitory postsynaptic currents (IPSCs), a high Cl⁻ intrapipette solution (120 mM KCl instead of K-gluconate) was used so that the reversal potential of [Cl⁻] was at -0 mV and GABA$_A$ receptor-mediated currents were inward at a holding potential of −60 mV. Light stimulation was delivered through the same objective via 10 ms pulses of blue laser (473 nm, FTEC2473-V65YF0, Blue Sky Research, Milpitas, USA). Viral infection in the OT was confirmed in brain slices during recordings. Pharmacological drugs CNO and SALB (dissolved in DMSO) were bath perfused.

### Immunohistochemistry

After patch clamp recordings, acute brain slices prepared form D1-Cre mice with AAV1-DIO-ChR2-EYFP virus and cholera toxin subunit B bilaterally injected into the OT and NAc, respectively, were immediately incubated in 4% paraformaldehyde (PFA) for 10–15 min, and then transferred into 1X phosphate buffered solution (PBS) overnight. Slices were washed with PBS three times (20 min each), and then incubated with 0.5% Triton X-100 in PBS for 10 min, followed by incubation with 0.5% Triton X-100 and 0.5% Tween-20 in PBS for 10 min. Slices were incubated with a primary antibody against tyrosine hydroxylase (TH) (1:500; Millipore, cat# AB152, host: rabbit) for 24 h at 4 °C. After three PBS washes (20 min each), the slices were incubated with a secondary antibody (1:1000; Life Technology, cat# A31573, host: donkey) and incubated for 24 h at 4 °C. Slices were washed three more times in 0.5% Triton X-100 and 0.5% Tween-20 in PBS (20 min each) and treated with glycerol in PBS (volume ratio 1:1) for 30 min followed by glycerol in PBS (volume ratio 7:3) for 30 min before being mounted onto superfrost slides for confocal imaging.

### Behavioral tests

All behavioral procedures were performed during the light cycle (9:00 am–12:00 pm) except for the sucrose preference test which was conducted during the dark cycle (6:00 pm–6:00 am).

All mice tested were transferred to the testing room 1 h before the test for habituation. The following behavioral tests were performed sequentially (stress level from low to high): open field test (OFT), light-dark box transition test (LDT), elevated-zero maze (EZM) test, forced swimming test (FST) and tail suspension test (TST) with an interval of 24 h between two individual behavioral assays. All apparatuses were cleaned with 70% ethanol before and between trials. Mice in all behavioral tests were videotaped using a webcam at 30 frames/sec, and behaviors were scored using ANY-maze 6.35 (Stoelting Co.) or manually by those who were blinded to the experimental conditions.

**Optogenetic and chemogenetic manipulations.** For optogenetic experiments, blue light (473 nm, 10–15 mW/mm², 20 Hz, 10 ms pulse, 10 s stimulation every 30 s) was applied to activate ChR2-expressing neurons after each daily CRS treatment for 15 min and during behavioral tests. Green light (532 nm), less efficient in activating these neurons, was applied with the same parameters as control. D3-Cre mice with AAV8-hSyn-DIO-hM3D(Gq)-mCherry (excitatory DREADD) and AAV8-hSyn-dF-HA-KORD-IRES-mCitrine (inhibitory DREADD) or AAV8-hSyn-DIO-mCherry (control virus) in the OT were intraperitoneally injected with saline or CNO (5 mg/kg) (for excitatory DREADD), or subcutaneously injected with DMSO or SALB (10 mg/kg) (for inhibitory DREADD), 30 min before behavioral tests.

**Open field test (OFT).** The mice were placed in an open field arena (40 cm × 40 cm) in a room with dim light and allowed to freely explore the apparatus for 5 min. The total distance traveled and the total time in the central zone (20 cm × 20 cm) were calculated. Orofacial grooming behavior was quantified during the OFT as previously

described[43]. In mice, a complete grooming bout consists of a syntactic chain that progresses sequentially from nose (phase I) to face (phase II) and head (phase III) grooming and ends with body licking (phase IV) and orofacial grooming contains phase I to III, but not IV. The beginning of an orofacial grooming bout was defined as when both paws were lifted to reach the face and the ending as when both paws returned to the cage floor.

**Light-dark box transition test (LDT).** The light-dark box (46 cm × 28 cm × 30 cm) was composed of two compartments. Two-thirds of the box was the light compartment and the remaining part was the dark compartment. Mice were placed in the center of the light box with the head oppositely facing the dark compartment, and were allowed to freely explore the two compartments for 10 min. The latency to the first entry into the dark compartment and the total time spent in the dark compartment were calculated.

**Elevated zero maze (EZM) test.** The zero-maze, composed of two open and two closed sections, was elevated 80 cm above the floor. The test mice were placed at the interface of an open and a closed section with the head facing the closed section. The latency to the first entry into the open section and the total time spent in the open sections were calculated.

**Forced swimming test (FST).** The FST, commonly used to assess antidepressant activity[83,84], was also used for testing depression-like behaviors (or "learned helplessness") in rodents[85,86]. Briefly, the mouse was placed in a cylindrical plexiglass tank (40 cm high × 12 cm in diameter) with water (25 ± 2 °C) in a depth of 10 cm. Mice underwent the swimming test for 6 min on the test day. The time spent in immobility was calculated post-hoc. Immediately after the FST, each mouse was removed from the water, towel-dried, and returned to its home cage. The water was changed and the cylinder was cleaned for each mouse tested.

**Tail suspension test (TST).** The TST is widely used for testing "behavioral despair", another depression-related symptom[87]. Briefly, the mouse, held by the tail, was suspended 50 cm from the floor for 6 min. The time spent in immobility was recorded. The mice were considered immobile only when they hung down passively and were completely motionless.

**Sucrose preference test.** Prior to the test, mice were habituated to the presence of two drinking bottles (one containing 2% sucrose and the other plain water) for 2 days in their home cage. Following this acclimation, mice had the free choice of either drinking the 2% sucrose solution or plain water for a period of 4 days. Water and sucrose solution intake was measured daily, and the positions of the two bottles were switched daily to reduce any confounding side bias. Sucrose preference was calculated as a percentage of the weight of sucrose intake over the total weight of fluid intake and averaged over the 4 days of testing. Some D3-Cre/ChR2 mice with optical fiber implantation in the OT were photostimulated for 15 min after each daily CRS treatment and during the tests with the same parameters as aforementioned. For chemogenetic manipulations, D3-Cre mice with AAV8-hSyn-DIO-hM3D(Gq)-mCherry and AAV8-hSyn-dF-HA-KORD-IRES-mCitrine in the OT were intraperitoneally (I.P.) injected with saline or CNO (5 mg/kg) (after two-week's CRS treatment), or subcutaneously injected with DMSO or SALB (10 mg/kg) (without CRS treatment), respectively, one time each day with only one drug in a counterbalanced way during the four days test session.

**Conditioned place preference (CPP).** A custom-built CPP apparatus consisted of a rectangular cage with three compartments: a left black chamber (35 cm × 20 cm) with a metal wire mesh floor, a connecting

zone (35 cm × 10 cm) with a smooth gray floor, and a right white chamber (35 cm × 20 cm) with a soft floor. The CPP test was conducted as previously described[88]. Briefly, the CPP test consisted of 4 days. Day 1 (pre-conditioning, 15 min): a preconditioning test was performed to obtain a baseline preference for the apparatus of each mouse tested. The side chamber a mouse spent the most time was assigned as the most preferred side, and the other side chamber as the least preferred side. On days 2 and 3 (conditioning): the mouse was firstly kept in either the most or least preferred side (counterbalanced across mice) for 15 min, then transferred to the other side for 15 min. Mice in the least-preferred side were paired with blue light stimulation (same parameters as aforementioned), and no photostimulations for mice in the most preferred side. On Day 4 (post-conditioning, 15 min): 24 h after the conditioning session on Day 3, the mice were placed back into the arena with all three compartments accessible, to evaluate preference for the stimulation and non-stimulation paired chambers. The CPP difference score was calculated by the time spent in the post-conditioning session minus that in the pre-conditioning session in the corresponding chamber.

**Rectification of potential grooming-related effects in behavioral tests.** Since blue light activation of OT D3 neurons induces orofacial grooming[43], it may influence behavioral measurements, including underestimation of the total distance traveled in the OFT and the immobility time in the FST and TST. We therefore rectified potential grooming-related deviations under light stimulation as follows. We first quantified the total duration of grooming in the 5 min OFT - approximately 11 s and 60 s grooming out of 100 s green and blue light stimulation, respectively. In the OFT, the rectified total distance traveled = the observed distance × 300 s/(300-11) s (for green light) or 300 s/(300-60) s (for blue light) because mice did not travel during the 11 s or 60 s grooming. The rectified time in the center zone = the observed time − light stimulation duration in the center zone × 11% (for green light) or 60% (for blue light). In the LDT, the rectified latency to the 1st entry into dark area = the observed time − light stimulation duration in light area × 11% (for green light) or 60% (for blue light). The rectified time in the dark area = the observed time − light stimulation duration in dark area × 11% (for green light) or 60% (for blue light). In the EZM, the rectified latency to the 1st entry into open section = the observed time − light stimulation duration in close section × 11% (for green light) or 60% (for blue light). The rectified time in open section = the observed time − light stimulation duration in open section × 11% (for green light) or 60% (for blue light). In the FST and TST, which lasted for 6 min (or 360 s) with 120 s light stimulation, we rectified the immobility time by adding "presumptive" grooming time ("presumptive" annotates the fact that mice could not groom in FST and TST) = 120 s × 11% (13.2 for green light) or 60% (72 s for blue light), extrapolated by the grooming time during light stimulation in OFT. Thus, the rectified immobility time = the observed immobility time + 13.2 s (for green light) or 72 s (for blue light).

Similarly, we rectified potential grooming-related deviations during behavioral tests in chemogenetic experiments. We quantified the increased grooming time by chemogenetic activation of OT D3 neurons in the OFT as 10 s out of 300 s (Fig. 6i). In the OFT, the rectified total distance traveled = the observed distance × 300 s/(300-10) s. For time in the center zone in the OFT, latency to the 1st entry into dark area and time in the dark area in LDT test, latency to the 1st entry into open section and time in open sections in the EZM test, the rectified time = the observed time*(1−10 s/300 s). In the FST and TST, the increased presumptive grooming time in 360 s = (360 s /300 s) × 10 s = 12 s. Thus, the rectified immobility time = the observed time + 12 s. In parallel, we quantified the decreased grooming time by chemogenetic inactivation of OT D3 neurons in the OFT as 18 s out of 300 s (Fig. 4i). In the OFT, the rectified total distance traveled = the observed distance × 300 s/(300 + 18) s. For time in the center zone in

the OFT, latency to the 1st entry into dark area and time in the dark area in LDT test, latency to the 1st entry into open section and time in open sections in the EZM test, the rectified time = the observed time *(1 + 18 s/300 s). In the FST and TST, the decreased presumptive grooming time in 360 s = 360 s/300 s × 18 s = 21.6 s. Thus, the rectified immobility time = the observed time − 21.6 s.

## Confocal imaging

Mice were deeply anesthetized (ketamine-xylazine; 200 and 20 mg/kg body weight, respectively) and perfused transcardially with 4% paraformaldehyde (PFA) in fresh phosphate buffered saline (PBS). The brain was post fixed in 4% PFA overnight at 4 °C and then transferred into PBS. Coronal slices (100 μm thick) were prepared using a Leica VT 1200S vibratome. The slices were treated with glycerol in PBS (volume ratio 1:1) for 30 min followed by glycerol in PBS (volume ratio 7:3) for 30 min before being mounted onto superfrost slides for imaging. Confocal imaging was performed by sequential scanning of slices at 10× and 40× in a SP5/Leica confocal microscope using LAS AF Lite.

## Statistical analysis

Shapiro–Wilk tests were used to verify normal distribution of datasets. For normally distributed datasets, parametric statistical tests (Student's $t$ test or two-way ANOVA test) were used; otherwise, non-parametric tests (Mann–Whitney or Wilcoxon matched-pairs signed rank test) were applied. Statistical analysis was performed in GraphPad Prism 7 and figures were assembled in Adobe Photoshop.

## Reporting summary

Further information on research design is available in the Nature Portfolio Reporting Summary linked to this article.

# Data availability

Due to multiple specialized platforms (including HEKA PULSE software, Igor Pro, ANY-maze, and LAS AF Lite) used to collect and/or analyze the data (ex vivo patch clamp recordings, behavioral videos, confocal images), it is not practical to make the raw data accessible to general readers. However, we will make the raw data available to readers upon request. Source data for all graphs are provided with this paper. Source data are provided with this paper.

# Code availability

All commercial software used to collect and analyze the data in this study are described. No custom code is used in this study.

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

## Acknowledgements

This work was supported by the National Institutes of Health R01NS117061 to M.M., D.W.W., and M.V.F., R01DA049545 and R01DA049449 to D.W.W. and M.M., R01DC006213 to M.M., the National Natural Science Foundation of China 82371515 to Y.F.Z., and the Talent Initiation BaiRen Plan Start-up Funds E251F811 to Y.F.Z.

## Author contributions

Conceptualization, Y.F.Z. and M.M.; Methodology, all authors; Investigation, Y.F.Z., J.W., Y.W., N.L.J., J.P.B., G.L., W.W., C.G., and H.S.; Formal Analysis, Data Curation, and Visualization, Y.F.Z.; Writing-Original Draft, Y.F.Z. and M.M.; Writing-Review & Editing, all authors; Supervision and Funding Acquisition, Y.F.Z., M.V.F., D.W.W., and M.M.

## Competing interests

The authors declare no competing interests.
