## [Peer Review File · Nature Communications]

Reviewers' Comments:

Reviewer #1:

Remarks to the Author:

This study is an interesting and elegant follow up on the previous work by this team reporting that the D3 receptor expressing cells in the Island of Calleja (IC) within the olfactory tuberculus. are activated during grooming and that optogenetic stimulation of these cells triggers bouts of grooming. In this study, the authors further developed the functional implications of this grooming behavior and the test the hypothesis that it could serve a therapeutic role under condition of chronic stress. The study uses chronic restrain stress (CRS) in mice to trigger a state of heightened anxiety-like and depressive-like behaviors, which are measured using light and dark box, elevated zero maze for anxiety and forced swim test and tail suspension for depression-like. Anhedonia is assessed with sucrose preference. The study first shows a decrease in the excitability of the D3 cells after the CRS and the induction of this depressive-like state. Then the study tests if inhibition or ablation of these neurons is sufficient to induce that state that replicates CRS and the answer is mixed with only increased immobility in the forced swim and tail suspension after ablation and more profound changes during inhibition.

The most interesting aspect of the study comes with the test of the main hypothesis and the potential therapeutic value. The study tests the effect of optogenetic stimulation and chemogenetic stimulation of D3 OT cells on reversing the effects of CRS. The main effects are a decrease in immobility during the forced swim test and the tail suspension when D3 OT cells are stimulated. These effects are very interesting and have potential therapeutic value. However, a concern is that the reduction on immobility is merely consequence of the increase in grooming that these manipulations cause. For example, activation or ablation of these neurons triggers bouts of grooming as it was shown in their previous work and here in Fig. 6 with chemogenetic activation.

1. Timing of the optogenetic stimulation. Currently the D3 cells are activated right after each restrain stress episode, which could test the hypothesis that grooming has a therapeutic value on its own and it could function as stress relief mechanism. However, cells are also activated during the test, which will induce grooming and affect the test and for example could explain the decrease in immobility. Question: is the opto stimulation during the test needed to effect of relieving the depressive-like phenotype? Or is the post-CRS stimulation sufficient?

2. Timing of the chemogenetic activation: Same here, can the effect also be obtained by CNO administration after each CRS event but not during the test?

3. CCP experiment is a nice addition and produces interesting data. But an alternative explanation of the increase time in the laser paired chamber after the conditioning sessions is just a reflection of increase exploration in a chamber in which mice did not get to explore during conditioning because of the continuous induction of grooming. The example traces hint at that difference. Can the authors plot distance traveled during the conditioning session to further explore or rule out this alternative interpretation? What other controls could be done? The collar experiment is an excellent one for probing the need of grooming, and shows that if the mice cannot groom, they won't show CCP, which will go along with this alternative interpretation of the data.

4. Fig. 2. Has Drd3 mRNA expression been tested in this Drd3-cre line? The D3-cre reporter mice does not seem to show fluorescence in the NAc in the coronal sections displayed. There have been reports of Drd3 expression in NAc, especially in D1-SPNs, as this could be an important matter when it comes to validation of the cre line. Can authors elaborate on this matter. The previous published work shows expression of reporter marked to be selective to IC in the OT and no expression in the NAc. However, is there data on Drd3 mRNA in these animals? This would be important to understand better the line and what is labeling, to serve as validation of this Drd3-cre line.

5. Fig.4 why is the total distance traveled of mice expressing inhibitory DREADDs so much lower than the other experiments (~ 5 vs 15-20 meters)? Can these reflect on the state state of the animals after DMSO injection?

6. More on Figure 4 controls. It is stated that two different controls were used in the distance traveled, tail suspension test and grooming. This control is SALB administration in mice that are NOT expressing KORD. Is the data combined and displayed in Figure 4, in addition to the Suppl. Figure 4? This is unclear. Please clarify.

7. There is no mention of Figure 4I, effect of inhibiting D3R neurons on grooming. please add. Same with Fig. 2. Ablation of D3 neurons in the OT. What happens with the grooming behavior? Based on previous work, the grooming is decreased. Can this be mentioned?

Reviewer #2:

Remarks to the Author:

Zhang et al. investigated the involvement of dopamine D3 receptor expressing neurons in the olfactory tubercle (OT) in "depression-like" behaviors. In the present manuscript, they report that 1) Chronic restraint stress (CRS) induces depression-like behaviors, measured by both FST and TST, in mice and decreases excitability of OT D3 neurons.

2) Ablation or inhibition of these neurons leads to depression-like behaviors, whereas activation of these neurons ameliorates CRS-induced depressive phenotypes.

3) Activation of OT D3 neurons has a rewarding effect, which diminishes if grooming is blocked.

4) OT D3 neurons inhibit OT SPNs that synapse onto VTA dopamine neurons that project to the NAc, presumably affect dopamine release into the NAc.

The strengths of the data lie in the consistent changes in immobility in both FST and TST after multiple manipulations (CRS, DTA mediated ablation of OT D3 neurons, chemogenetic inhibition and activation of OT D3 neurons, optogenetic activation of OT D3 neurons), and that those changes are correlated nicely with changes in OT D3 neuron excitability. Those data fit with their main hypothesis. The discovery of the new pathway (point 4 above) in modulating the reward pathway is also very significant. These data also support this idea that the reward pathway is associated with "depression-like" behaviors, although the literatures on the reward pathway and on "depression-like" behaviors almost belong to two different fields. The observations here offer a unique opportunity to unite these two fields.

The biggest weakness is that the data on "anxiety" tests and on sucrose preference test show changes in almost all directions after various manipulations. The inconsistency is not well explained, at least the explanations in the manuscript are somewhat arbitrary without strong premises. Furthermore, multiple comparisons were used often without corrections for potential Type I errors. This is especially problematic with all kinds of changes that do not fit with a simple hypothesis. Those data complicate the otherwise straightforward story about the reward pathway and "depression-like" behaviors.

One technical issue in experimental design: In CPP pre-conditioning sessions, the least preferred chamber was selected to be the "rewarded" chamber during conditioning. Such a design could artificially cause apparent CPP due to "regression to the mean", a common statistical phenomenon.

Minor issues:

One minor conceptual weakness is that the role of grooming does not fit well into the model. The overall model is a "circuit" level model. It's not clear how this "behavior" could fit in the middle of the circuit model to "mediate" reward or reduced depression. Furthermore, CRS mice exhibit more grooming than controls.

Olfactory bulbectomy is one of the commonly used depression models. It induces behavioral deficits similar to clinical depression. It is not even discussed in the manuscript.

It is possible that the FST and TST are more related to learning and/or motivation in which the

reward pathway plays the central role. The sucrose preference test is more related to hedonia which seems to have different mechanisms based on the literature.

BAC transgenic mice often do not match the endogenous expression pattern. It's not clear if the D3-Cre mice were validated.

Reviewer #3:

Remarks to the Author:

The primary goal of the current paper was to characterize the role of olfactory tubercle (OT) dopamine D3 receptor-expressing granule cells in reward and motivation-related behaviors. Prior research has shown that OT D3 neurons bidirectionally regulate self-grooming in mice, a repetitive behavior known to reduce stress. Given that self-grooming also increases levels of dopamine in the nucleus accumbens, it is possible that OT D3 neurons mediate reward and affective states. The authors of the paper interrogated the role of OT D3 neurons with a variety of technical approaches. Specifically, they combined both optogenetic and chemogenetic approaches, genetic ablation of OT D3 neurons, electrophysiology, and a diverse a range of behavioral assays to understand the relationship between OT D3 and reward and motivation. Through a series of both loss-of-function and gain-of-function experiments, researchers highlight a bidirectional role for OT D3 neurons in mediating affective behaviors.

Strengths

- The authors were thorough and used an assortment of experimental techniques to probe the effects OT D3 loss-of-function and gain-of-function in the context of chronic restraint stress.
- The authors included both males and females in the study.
- The study provides new and significant information into the role of specific OT cell types mediating affective behavioral states.

Weaknesses

1. The authors use 'anxiety-like' and 'depression-like' verbiage throughout the paper and in some instances state "depression phenotypes". The authors should be careful with these descriptions. The latter term anthropomorphizes mice. Further, FST and TST have no human equivalent behavioral phenotype in neuropsychiatric disorders and the field is shifting away from these behaviors. It is suggested that they use more specific language to describe their behavioral paradigms and use terms like affective behaviors.
2. In Figure 2 it is not clear why the authors combined D1 and D2 expressing SPNs. Since they are separate neuron populations it would be appropriate to analyze these cell types separately.
3. Despite the breadth of downstream behavioral experiments, there was a missed opportunity to do a Splash Test experiment given the emphasis on OT D3 mediated self-grooming. Perhaps orofacial grooming (as described in the paper) is unique from overall grooming?
4. In Figure 3A it would be helpful to include a high power image of the virus labeling along with the current images. While there is clearly less Chr2-EYFP suggestions ablation of OT D3, a higher power image would allow visualization of the cells would better help the reader to visualize the effectiveness of the ablation.
5. Please include more methodological context regarding the self-grooming experiment and provide quantification. The self-grooming experiment is not included in any of the timelines and no additional information regarding context (when, where, how) is provided.

We greatly appreciate the positive assessment and constructive feedback from all three reviewers. The original comments are in *italic* and our responses in purple. The revised text in the manuscript is in red.

Reviewer #1 (Remarks to the Author):

This study is an interesting and elegant follow up on the previous work by this team reporting that the D3 receptor expressing cells in the Island of Calleja (IC) within the olfactory tuberculous. are activated during grooming and that optogenetic stimulation of these cells triggers bouts of grooming. In this study, the authors further developed the functional implications of this grooming behavior and the test the hypothesis that it could serve a therapeutic role under condition of chronic stress. The study uses chronic restrain stress (CRS) in mice to trigger a state of heightened anxiety-like and depressive-like behaviors, which are measured using light and dark box, elevated zero maze for anxiety and forced swim test and tail suspension for depression-like. Anhedonia is assessed with sucrose preference. The study first shows a decrease in the excitability of the D3 cells after the CRS and the induction of this depressive-like state. Then the study tests if inhibition or ablation of these neurons is sufficient to induce that state that replicates CRS and the answer is mixed with only increased immobility in the forced swim and tail suspension after ablation and more profound changes during inhibition.

The most interesting aspect of the study comes with the test of the main hypothesis and the potential therapeutic value. The study tests the effect of optogenetic stimulation and chemogenetic stimulation of D3 OT cells on reversing the effects of CRS. The main effects are a decrease in immobility during the forced swim test and the tail suspension when D3 OT cells are stimulated. These effects are very interesting and have potential therapeutic value.

However, a concern is that the reduction on immobility is merely consequence of the increase in grooming that these manipulations cause. For example, activation or ablation of these neurons triggers bouts of grooming as it was shown in their previous work and here in Fig. 6 with chemogenetic activation..

Response: We agree with the reviewer on this important point. Optogenetic activation of OT D3 neurons increased self-grooming, potentially deviating behavioral measurements, including underestimation of the immobility time in the forced swimming test (FST) and tail suspension test (TST). We therefore rectified the observed measurements by correcting potential grooming-related deviations. In the FST and TST, which lasted for 6 min (or 360 s) with 120 s light stimulation, we corrected the observed immobility time by adding “presumptive” grooming time during light stimulation (“presumptive” annotates the fact that mice could not groom in FST and TST). The presumptive grooming time was extrapolated by the grooming time during light stimulation in the open field test (OFT). The total grooming time during 100 s green or blue stimulation in the 5 min (or 300 s) OFT was approximately 11 s (or 11% of the stimulation) or 60 s (or 60% of the stimulation), respectively. In the FST and TST, the presumptive grooming time in 120 s light stimulation = 120 s x 11% (13.2 s for green light) or 60% (72 s for blue light). Thus, the rectified immobility time = the observed immobility time + 13.2 s (for green light) or 72 s (for blue light). As detailed in the Results (Page 7-9) and Methods (Page 21-22), we rectified potential grooming-related deviations in optogenetic and chemogenetic experiments for all the behavioral tests. Same conclusions were reached for the FST and TST (c.f., new Supplemental Figs. S5, S6, S9 to Figs. 4, 5, 6, respectively).

1. Timing of the optogenetic stimulation. Currently the D3 cells are activated right after each restraint stress episode, which could test the hypothesis that grooming has a therapeutic value on its own and it could function as stress relief mechanism. However, cells are also activated during the test, which will induce grooming and affect the test and for example could explain the decrease in immobility. Question: is the opto stimulation during the test needed to effect of relieving the depressive-like phenotype? Or is the post-CRS stimulation sufficient?

Response: We thank the reviewer for pointing this out. We conducted a new set of experiments in which we optogenetically activated OT D3 neurons only after each CRS session but not during the behavioral tests. This manipulation failed to reverse CRS-induced depression-like behaviors in the FST and TST (Supplemental Fig. S7). The main text is revised accordingly on Page 8.

2. Timing of the chemogenetic activation: Same here, can the effect also be obtained by CNO administration after each CRS event but not during the test?

Response: As stated in our Response to question #1 above, optogenetic activation of OT D3 neurons only after each CRS session was not effective in relieving CRS-induced depression-like behaviors (new Supplemental Fig. S7). We initially showed that optogenetic activation of D3 neurons activation after each CRS session and during the tests together reversed CRS-induced behavioral changes (Fig. 5). These findings strongly suggest that activation of D3 neurons during the behavioral tests is required for the relieving effect. This notion is further supported by the finding that chemogenetic activation of D3 neurons only during the behavioral tests was sufficient (Fig. 6). Taken together, these results suggest that chemogenetic activation (CNO administration) only after each CRS session is unlikely effective. The main text is revised accordingly on Page 8.

3. CCP experiment is a nice addition and produces interesting data. But an alternative explanation of the increase time in the laser paired chamber after the conditioning sessions is just a reflection of increase exploration in a chamber in which mice did not get to explore during conditioning because of the continuous induction of grooming. The example traces hint at that difference. Can the authors plot distance traveled during the conditioning session to further explore or rule out this alternative interpretation? What other controls could be done? The collar experiment is an excellent one for probing the need of grooming, and shows that if the mice cannot groom, they won't show CCP, which will go along with this alternative interpretation of the data.

Response: We appreciate the reviewer's point. We compared the total distance travelled in the blue laser-paired chamber during the 2nd conditioning session between D3-Cre control and D3-Cre/ChR2 mice and they were very similar (rebuttal rFig. 1). It is curious that blue light stimulated D3-Cre/ChR2 mice did not show reduced total distance travelled even they groomed more. This is likely attributed to the fact that blue light was only delivered in 1/3 of the time (10 sec ON and 20 sec OFF; 10 sec ON led to an average of 6 sec grooming in the open field) and a mouse spent $(1/3) \times (6/10) = 19.8\%$ of time grooming in the light-paired chamber during a conditioning session. This suggests that the reduction of 19.8% of travel time did not lead to a significant shorter distance travelled, consistent with the measurements in the open field (Figs. 5 and S7). Furthermore, the total distance traveled in the post-conditioning session was also similar between D3-Cre control and D3-Cre/ChR2 mice (rFig. 1). Therefore, the increased exploration time in the blue laser-paired chamber is unlikely due to the instinct

of mice preferring to explore the novel environment/area, but rather due to OT D3 neuron activation/grooming-induced reward effects.

rFig. 1. Total distance travelled during the conditioning session in the blue laser-paired chamber and during post-conditioning session in the entire arena in CPP test. No significant difference was observed in both D3-Cre and D3-Cre/ChR2 mice. n = 8 mice per group. Data are expressed as mean \pm SEM.

4. Fig. 2. Has *Drd3* mRNA expression been tested in this *Drd3-cre* line? The *D3-cre* reporter mice does not seem to show fluorescence in the NAc in the coronal sections displayed. There have been reports of *Drd3* expression in NAc, especially in D1-SPNs, as this could be an important matter when it comes to validation of the *cre* line. Can authors elaborate on this matter. The previous published work shows expression of reporter marked to be selective to IC in the OT and no expression in the NAc. However, is there data on *Drd3* mRNA in these animals? This would be important to understand better the line and what is labeling, to serve as validation of this *Drd3-cre* line.

Response: *Drd3* is indeed expressed broadly in the ventral striatum including the islands of Calleja and NAc (see the *in situ* hybridization data of *Drd3* gene in the Allen Brain Atlas: <https://mouse.brain-map.org/experiment/show/75038431>). There are two available D3-Cre transgenic mouse lines, the B6.FVB(Cg)-Tg(*Drd3-cre*)KI196Gsat/Mmucd and B6.FVB(Cg)-Tg(*Drd3-cre*)KI198Gsat/Mmucd, and the detailed characterization of Cre expression for each line is available in Allen Brain Atlas Transgenic Characterization (for KI196 line: <https://connectivity.brain-map.org/transgenic/experiment/286328839>; for KI198 line: <https://connectivity.brain-map.org/transgenic/experiment/304166273>). The main difference between the two lines is that the Cre expression in the KI198 line is more restricted to the islands of Calleja, which is our major target. In our previous publication (Zhang et al 2021), we quantified D3-Cre/tdTomato neurons in the OT, NAc and ventral pallidum, which account for 83.3%, 11.3% and 5.4% of the total tdTomato+ neurons in this region, respectively. Within the OT, although there are some ‘loose’ D3-Cre/tdTomato neurons, the vast majority (~90%) of OT D3-Cre/tdTomato neurons can be categorized as ‘dense’ clusters belonging to the islands of Calleja. We added the following sentence in the Methods on Page 15: “This D3-Cre line

was chosen because Cre expression is more restricted to the islands of Calleja neurons” by citing our previous publication.

5. Fig.4 why is the total distance traveled of mice expressing inhibitory DREADDs so much lower than the other experiments (~ 5 vs 15-20 meters)? Can these reflect on the state of the animals after DMSO injection?

Response: We noticed that mice treated with DMSO had a reduced total distance travelled compared to other control groups. DMSO is known to influence animal behaviors presumably due to its toxicity (e.g., Colucci et al 2008). Since each experimental group has its own DMSO control, the difference between DMSO and SALB should reflect the effect of SALB-mediated action on KORD (Fig. 4). This is mentioned on Page 7.

6. More on Figure 4 controls. It is stated that two different controls were used in the distance traveled, tail suspension test and grooming. This control is SALB administration in mice that are NOT expressing KORD. Is the data combined and displayed in Figure 4, in addition to the Suppl. Figure 4? This is unclear. Please clarify.

Response: We are sorry for the confusion. Data in Fig. 4 (with KORD) and Supplemental Fig. S4 (without KORD) are collected from independent experiments using two different cohorts of mice and each group has its own DMSO control. This is clarified on Page 7. “To exclude the potential non-specific effect of SALB, we bilaterally injected a control virus without DREADD (Cre-dependent AAV8-DIO-mCherry) into the OT of another cohort of D3-Cre mice. Application of DMSO or SALB did not change behaviors in these control mice (Supplemental Fig. S4), supporting that inhibitory DREADD-induced effects result from the interaction between SALB and KORD.”

7. There is no mention of Figure 4I, effect of inhibiting D3R neurons on grooming. please add. Same with Fig. 3. Ablation of D3 neurons in the OT. What happens with the grooming behavior? Based on previous work, the grooming is decreased. Can this be mentioned?

Response: In the original submission, Fig. 4I was only mentioned later with Fig. 6I. In the revision, we added the effect of inhibiting D3 neurons on grooming on Page 7: “Moreover, chemogenetic inhibition of OT D3 neurons decreased grooming behavior (Fig. 4I), which potentially influences the behavioral measurements, including overestimation of the immobility time in the FST and TST. We therefore rectified potential grooming-related deviations upon chemogenetic inhibition (detailed in Methods) and similar conclusions could be drawn (Supplemental Fig. S5A-E).” For the effects of ablation of D3 neurons on grooming, we quantified the detailed changes in our previous publication and stated on Page 6: “As we previously reported³⁴, ablation of OT D3 neurons significantly reduced the total grooming time.”

Reviewer #2 (Remarks to the Author):

Zhang et al. investigated the involvement of dopamine D3 receptor expressing neurons in the olfactory tubercle (OT) in “depression-like” behaviors. In the present manuscript, they report that 1) Chronic restraint stress (CRS) induces depression-like behaviors, measured by both FST and TST, in mice and decreases excitability of OT D3 neurons.

2) Ablation or inhibition of these neurons leads to depression-like behaviors, whereas activation of these neurons ameliorates CRS-induced depressive phenotypes.

3) Activation of OT D3 neurons has a rewarding effect, which diminishes if grooming is blocked.

4) OT D3 neurons inhibit OT SPNs that synapse onto VTA dopamine neurons that project to the NAc, presumably affect dopamine release into the NAc.

The strengths of the data lie in the consistent changes in immobility in both FST and TST after multiple manipulations (CRS, DTA mediated ablation of OT D3 neurons, chemogenetic inhibition and activation of OT D3 neurons, optogenetic activation of OT D3 neurons), and that those changes are correlated nicely with changes in OT D3 neuron excitability. Those data fit with their main hypothesis. The discovery of the new pathway (point 4 above) in modulating the reward pathway is also very significant. These data also support this idea that the reward pathway is associated with “depression-like” behaviors, although the literatures on the reward pathway and on “depression-like” behaviors almost belong to two different fields. The observations here offer a unique opportunity to unite these two fields.

The biggest weakness is that the data on “anxiety” tests and on sucrose preference test show changes in almost all directions after various manipulations. The inconsistency is not well explained, at least the explanations in the manuscript are somewhat arbitrary without strong premises. Furthermore, multiple comparisons were used often without corrections for potential Type I errors. This is especially problematic with all kinds of changes that do not fit with a simple hypothesis. Those data complicate the otherwise straightforward story about the reward pathway and “depression-like” behaviors.

Response: We thank the reviewer for raising this point and for suggesting corrections for potential Type I errors. We have redone the statistical analysis on anxiety-like behaviors by applying the conservative Bonferroni correction method (Figs. 1, 3-6 and Supplemental Figs. S1, S2, S4-S9). After correction, only a few tests showed significant differences, including the elevated zero maze test after CRS (Fig. 1D) and the light-box test after chemogenetic inhibition of OT D3 neurons (Fig. 4D and Supplemental Fig. 5D after rectification of grooming-related activity), suggesting that CRS-induced anxiety-like behaviors may engage neural circuits in addition to OT D3 neurons. We expanded the discussions on these points on Page 11-12: “Unlike the consistent performance in depression-like behavior assays (FST and TST), CRS mice exhibited divergent states in different anxiety assays. Anxiety-like behaviors were observed in the EZM test, but not in the OFT and LDT test (Fig. 1B-D), suggesting that these behavioral assays may have different sensitivities or test different aspects of anxiety. Given that there is no single ideal mouse test for anxiety and that each existing test has its advantages⁵⁵, a combination of different behavioral tests produces a better understanding in anxiety-related processes⁵⁶. Our findings further support the necessity of using multiple behavioral assays for testing anxiety-like behaviors.” Also on Page 13: “Curiously, the anxiety-like behaviors induced by CRS (only in the EZM test) and chemogenetic inhibition of OT D3 neurons (only in the LDT test) are different, suggesting that CRS-induced anxiety-like behaviors engage neural circuits in addition to OT D3 neurons.”

One technical issue in experimental design: In CPP pre-conditioning sessions, the least preferred chamber was selected to be the “rewarded” chamber during conditioning. Such a design could artificially cause apparent CPP due to “regression to the mean”, a common statistical phenomenon.

Response: In the published CPP studies, it is a common strategy to designate the least preferred chamber as the stimulus paired chamber (e.g., Calcagnetti & Schechter 1994, Frances et al 2004, Imaizumi et al 2001, Wahis et al 2021), so that the rewarding effect should not be due to the baseline

preference of that chamber. This ‘biased’ CPP design has been utilized in hundreds of studies which have yielded replicable data regarding reinforcing properties of stimuli ranging from psychoactive substances to social partners and that was an additional rationale for our selection.

Minor issues:

One minor conceptual weakness is that the role of grooming does not fit well into the model. The overall model is a “circuit” level model. It’s not clear how this “behavior” could fit in the middle of the circuit model to “mediate” reward or reduced depression. Furthermore, CRS mice exhibit more grooming than controls.

Response: We appreciate this point. The grooming behavior is not simply fit into the circuit model for two reasons: (1) there are multiple brain regions and cell types (including the OT D3 neurons) mediating this behavior, and (2) the relationship between grooming and affective behaviors is also multifaceted. We expanded our discussion on Page 13-14. “Interestingly, CRS treatment decreased excitability of OT D3 neurons (Fig. 2), which should act to reduce the grooming drive from these neurons. However, CRS mice exhibited more grooming than controls (Fig. 1H), suggesting that other grooming centers may be more active after CRS treatment to ensure this adaptive behavior in stressed situation^{35, 37, 44, 47}.”

Olfactory bulbectomy is one of the commonly used depression models. It induces behavioral deficits similar to clinical depression. It is not even discussed in the manuscript.

Response: We thank the reviewer for this suggestion. We discussed the olfactory bulbectomy on Page 12: “Notably, olfactory bulbectomy in rodents lead to depression-like behaviors⁵⁷, but the underlying mechanisms are not fully understood. Since the OT receives direct inputs from the OB, it would be interesting to determine whether OT circuitry contributes to depression-like behaviors in bulbectomized rodents.”

It is possible that the FST and TST are more related to learning and/or motivation in which the reward pathway plays the central role. The sucrose preference test is more related to hedonia which seems to have different mechanisms based on the literature.

Response: We agree with the reviewer and have included the following discussion on Page 12-13. “One possibility is that different aspects of CRS-induced changes in affective behaviors are mediated by distinct neuronal types and/or brain regions, which is supported by the finding that distinct ventral pallidal neurons mediate separate depression-like behaviors⁵⁸. Activation of OT D3 neurons may specifically improve “the decreased motivation for activity” in CRS mice, while anhedonic phenotypes might be mediated by other neuronal subtypes such as cholinergic neurons^{25, 59} and/or other brain regions.”

BAC transgenic mice often do not match the endogenous expression pattern. It’s not clear if the D3-Cre mice were validated.

Response: Please see our response to comment #4 by Reviewer 1.

Reviewer #3 (Remarks to the Author):

The primary goal of the current paper was to characterize the role of olfactory tubercle (OT) dopamine D3 receptor-expressing granule cells in reward and motivation-related behaviors. Prior research has shown that OT D3 neurons bidirectionally regulate self-grooming in mice, a repetitive behavior known to reduce stress. Given that self-grooming also increases levels of dopamine in the nucleus accumbens, it is possible that OT D3 neurons mediate reward and affective states. The authors of the paper interrogated the role of OT D3 neurons with a variety of technical approaches. Specifically, they combined both optogenetic and chemogenetic approaches, genetic ablation of OT D3 neurons, electrophysiology, and a diverse range of behavioral assays to understand the relationship between OT D3 and reward and motivation. Through a series of both loss-of-function and gain-of-function experiments, researchers highlight a bidirectional role for OT D3 neurons in mediating affective behaviors.

Strengths

- The authors were thorough and used an assortment of experimental techniques to probe the effects OT D3 loss-of-function and gain-of-function in the context of chronic restraint stress.*
- The authors included both males and females in the study.*
- The study provides new and significant information into the role of specific OT cell types mediating affective behavioral states.*

Weaknesses

1. The authors use ‘anxiety-like’ and ‘depression-like’ verbiage throughout the paper and in some instances state “depression phenotypes”. The authors should be careful with these descriptions. The latter term anthropomorphizes mice. Further, FST and TST have no human equivalent behavioral phenotype in neuropsychiatric disorders and the field is shifting away from these behaviors. It is suggested that they use more specific language to describe their behavioral paradigms and use terms like affective behaviors.

Response: We appreciate the reviewer’s comment and suggestion. In the revised manuscript, we used “affective behaviors” to replace “anxiety- and depression-like behaviors” wherever possible and removed phrases like depression phenotypes for mice. However, we kept “anxiety-like” and “depression-like” behaviors when we need to describe specific ‘phenotypically-relevant’ tests (OFT, LDT and EZM versus FST and TST).

2. In Figure 2 it is not clear why the authors combined D1 and D2 expressing SPNs. Since they are separate neuron populations it would be appropriate to analyze these cell types separately.

Response: We thank the reviewer’s suggestion. In the revised Fig. 2C, 2D, we separated D1 and D2 SPNs. The main text and figure legend are revised accordingly (Page 5).

3. Despite the breadth of downstream behavioral experiments, there was a missed opportunity to do a Splash Test experiment given the emphasis on OT D3 mediated self-grooming. Perhaps orofacial grooming (as described in the paper) is unique from overall grooming?

Response: We thank the reviewer for this suggestion. In CRS-treated D3-Cre/ChR2 mice, we applied optogenetic stimulations to OT D3 neurons when mice were performing body licking during the splash test (10% sucrose solution sprayed on the dorsal coat of a mouse). Consistent with our previous report (Zhang et al., 2021), activation of OT D3 neurons by blue light terminated body licking in 76% of the trials and also initiated orofacial grooming in 65% of the trials, in sharp contrast to green light (rFig. 2). We therefore did not use this test to assess “motivation” as a readout for “depression-like” behaviors.

rFig. 2. Orofacial grooming induced by optogenetic activation of OT D3 neurons suppresses sucrose splash-triggered body licking. Blue light (activation of D3-Cre/ChR2 neurons; 10 ms pulses, 20 Hz for 10 s) or green light (same parameters as blue light as comparison) was delivered when sucrose splash-triggered body licking occurred (n = 5 mice). Left, percent of trials in which body licking terminated. Right, percent of trials in which body licking terminated and orofacial grooming initiated. Each mouse was tested for 20 trials. Data are expressed as mean ± SEM. Wilcoxon signed-rank test. $p < 0.0001$ for (D) and $p = 0.0005$ for (E). *** $p < 0.001$, **** $p < 0.0001$.

4. In Figure 3A it would be helpful to include a high power image of the virus labeling along with the current images. While there is clearly less ChR2-EYFP suggestions ablation of OT D3, a higher power image would allow visualization of the cells would better help the reader to visualize the effectiveness of the ablation.

Response: We appreciate the reviewer’s suggestion. We have added high magnification images in Figure 3A and revised the figure legend accordingly.

5. Please include more methodological context regarding the self-grooming experiment and provide quantification. The self-grooming experiment is not included in any of the timelines and no additional information regarding context (when, where, how) is provided.

Response: This was stated in the figure legends of Figs. 1, 4, 6, S1 and S2: “Total grooming time in the open field test...”. In the revision, we added the following information in the Methods. On Page 19: “Orofacial grooming behavior was quantified during the OFT as previously described³⁴: the beginning

of a grooming bout was defined as when both paws were lifted to reach the face and the ending as when both paws returned to the cage floor.”

References

- Calcagnetti DJ, Schechter MD. 1994. Nicotine place preference using the biased method of conditioning. *Prog Neuropsychopharmacol Biol Psychiatry* 18: 925-33
- Colucci M, Maione F, Bonito MC, Piscopo A, Di Giannuario A, Pieretti S. 2008. New insights of dimethyl sulphoxide effects (DMSO) on experimental in vivo models of nociception and inflammation. *Pharmacol Res* 57: 419-25
- Frances H, Le Foll B, Diaz J, Smirnova M, Sokoloff P. 2004. Role of DRD3 in morphine-induced conditioned place preference using drd3-knockout mice. *Neuroreport* 15: 2245-9
- Imaizumi M, Takeda M, Sawano S, Fushiki T. 2001. Opioidergic contribution to conditioned place preference induced by corn oil in mice. *Behav Brain Res* 121: 129-36
- Wahis J, Baudon A, Althammer F, Kerspern D, Goyon S, et al. 2021. Astrocytes mediate the effect of oxytocin in the central amygdala on neuronal activity and affective states in rodents. *Nat Neurosci* 24: 529-41
- Zhang YF, Vargas Cifuentes L, Wright KN, Bhattarai JP, Mohrhardt J, et al. 2021. Ventral striatal islands of Calleja neurons control grooming in mice. *Nat Neurosci* 24: 1699-710

Reviewers' Comments:

Reviewer #2:

Remarks to the Author:

The authors addressed all my concerns except the following remaining issue:

"In CPP pre-conditioning sessions, the least preferred chamber was selected to be the "rewarded" chamber during conditioning. Such a design could artificially cause apparent CPP due to "regression to the mean", a common statistical phenomenon."

The authors responded:

"In the published CPP studies, it is a common strategy to designate the least preferred chamber as the stimulus paired chamber (e.g., Calcagnetti & Schechter 1994, Frances et al 2004, Imaizumi et al 2001, Wahis et al 2021), so that the rewarding effect should not be due to the baseline preference of that chamber. This 'biased' CPP design has been utilized in hundreds of studies which have yielded replicable data regarding reinforcing properties of stimuli ranging from psychoactive substances to social partners and that was an additional rationale for our selection."

This design is certainly problematic and "regression to the mean" is a real problem. However, I agree with the authors that many papers have been published with this approach. This does not mean that this approach is correct, yet the authors followed the convention. I suggest that the authors could perform a secondary analysis to ensure that the observed CPP is real. I will let the editor(s) decide if this is necessary. I do not need to review it again.

Reviewer #3:

Remarks to the Author:

The authors have addressed all concerns!

The original comments are in *italic* and our responses in purple.

Reviewer #2 (Remarks to the Author):

The authors addressed all my concerns except the following remaining issue:

"In CPP pre-conditioning sessions, the least preferred chamber was selected to be the "rewarded" chamber during conditioning. Such a design could artificially cause apparent CPP due to "regression to the mean", a common statistical phenomenon."

The authors responded:

"In the published CPP studies, it is a common strategy to designate the least preferred chamber as the stimulus paired chamber (e.g., Calcagnetti & Schechter 1994, Frances et al 2004, Imaizumi et al 2001, Wahis et al 2021), so that the rewarding effect should not be due to the baseline preference of that chamber. This 'biased' CPP design has been utilized in hundreds of studies which have yielded replicable data regarding reinforcing properties of stimuli ranging from psychoactive substances to social partners and that was an additional rationale for our selection."

This design is certainly problematic and "regression to the mean" is a real problem. However, I agree with the authors that many papers have been published with this approach. This does not mean that this approach is correct, yet the authors followed the convention. I suggest that the authors could perform a secondary analysis to ensure that the observed CPP is real. I will let the editor(s) decide if this is necessary. I do not need to review it again.

Response: We appreciate this point raised by the reviewer. We used the commonly adapted CPP test which assigns the least preferred side as the conditioning side to rule out the possibility that post-conditioning preference is due to baseline preference. This design may have the potential limitation due to "regression to the mean" as pointed out by the reviewer. However, this effect unlikely explains the CPP showed by D3-Cre/ChR2 mice after blue light conditioning since control D3-Cre mice maintained their preference between the post- and pre-conditioning session in the same design (Fig. 7B). We believe that our conclusion is well supported without a secondary analysis. This point is discussed on Page 9.

Reviewer #3 (Remarks to the Author):

The authors have addressed all concerns!

Response: We appreciate the positive assessment.